# Regulating Internal Alignment Flows for Robust Learning Under Spurious Correlations

**Rajeev Ranjan Dwivedi**[*]**, Mohammedkaif Kalagond**[*]**, Niramay M. Patel & Vinod K Kurmi**
Indian Institute of Science Education and Research Bhopal (IISERB)
{`rajeev22, mohammedkaif23, niramay23, vinodkk`}@iiserb.ac.in

## Abstract

Deep models often exploit spurious correlations (e.g., backgrounds or dataset artifacts), hurting worst-group performance. We propose **Alignment-Gated Suppression (AGS)**, a lightweight, plug-in regularizer that intervenes inside the network during training. AGS tracks a class-conditional, confidence-weighted contribution for each neuron (more negative $\Leftrightarrow$ stronger support) and applies a percentile-based, multiplicative decay to the most extreme contributors, reducing overconfident shortcut pathways while leaving other features relatively more influential. AGS integrates with standard ERM, requires no group labels, and adds $< 5\%$ training overhead. We provide analysis linking AGS to minority-margin gains, path-norm-like capacity control, and stability benefits via EMA-smoothed gating. Empirically, AGS improves worst-group accuracy and calibration vs. ERM and is competitive with state-of-the-art methods across spurious-correlation benchmarks (e.g., Waterbirds, CelebA, BAR, COCO), while maintaining strong average accuracy. These results suggest that regulating internal alignment flow is a simple and scalable route to robustness without group labels.

## 1 Introduction

Deep learning models have achieved remarkable success across a wide range of tasks, establishing them as the foundation of modern AI. However, understanding how these models achieve such results reveals critical flaws in their learning process. A critical issue is that such models often rely on spurious correlations superficial features like backgrounds, attributes, or dataset artifacts rather than the core cues (You et al., 2025; Beery et al., 2018). While this shortcut learning can produce high average-case accuracy, it can also cause catastrophic failures on minority or worst-case subpopulations, where these spurious cues no longer hold (Geirhos et al., 2020; Yang et al., 2023; Hashimoto et al., 2018; Tatman, 2017; Duchi et al., 2019). Such vulnerabilities make models unreliable in real-world deployments, especially in safety- and fairness-critical applications. Addressing this challenge requires encouraging models to focus their predictions on robust, transferable features rather than shortcuts, with the goal of improving worst-group accuracy and narrowing group disparities while maintaining strong overall performance (Izmailov et al., 2022).

A large body of robust learning methods attacks this problem by modifying data (e.g., group-aware reweighting Nam et al. (2020); Kim et al. (2019); Lee et al. (2022), augmentations Kim et al. (2021)) or objectives that require access to group or environment labels (e.g., DRO-style surrogates and invariance constraints (Sagawa et al., 2020)). These approaches can be highly effective when high-quality group annotations and multiple environments are available, but in many real deployments such side information is missing, expensive, or unreliable Mehrabi et al. (2021). Moreover, interventions that operate outside the network (data-level) or only at the loss often lack direct control over the internal contributors of specific neurons or connections that propagate spurious alignment energy forward Dwivedi et al. (2024); K Kurmi et al. (2022).

We take a complementary path and intervene inside the model during training. We introduce *Alignment-Gated Suppression* (AGS), a plug-and-play, group-agnostic regularizer that selectively suppresses spurious feature neurons using a class-conditional, softmax-weighted measure of on-batch

---

[*]These authors contributed equally to this work.

alignment energy. Intuitively, for each class, we estimate how much each neuron contributes to class-consistent predictions; neurons whose smoothed alignment score falls in the most negative tail are multiplicatively shrunk during training, while others remain unchanged except for a mild global decay applied for stability. The mechanism integrates seamlessly with standard cross-entropy optimization, requires no architectural changes, and adds less than 5% training-time overhead.

Concretely, AGS maintains an exponential moving average (EMA) of per-neuron, per-class alignment energy computed from mini-batches, and applies a percentile-gated decay to connections whose smoothed alignment score falls below a data-driven threshold. More negative alignment score indicates stronger class-alignment, and suppressing the extreme negative tail prevents over-confident shortcut-driven pathways, leaving the remaining connections relatively more influential. This alignment-gated suppression thus down-weights neurons tied to spurious cues while indirectly encouraging reliance on more robust features, yielding representations that improve generalization and benefit minority groups even without group annotations.

Beyond empirical gains, our method is also clearly explained through theory. If we look at the model from the perspective of its final linear classifier (assuming the features it receives are well-behaved), the update we apply has three main effects: ❶ increases margins for hard or minority groups by shrinking connections that rely on spurious cues; ❷ controls model capacity in a way similar to path-norm regularization, by reducing the combined strength of weights and activations; ❸ improves algorithmic stability due to EMA smoothing and mild decay, which in turn connects to calibration benefits by reducing overconfident logits from misaligned neurons. These properties explain why the method tends to lift average accuracy, worst-group accuracy and lower calibration error. Empirically, across standard spurious-correlation benchmarks (Waterbirds, CelebA, COCO, BAR) our method consistently improves average and worst-group performance while incurring minimal training overhead. Our analyses further illustrate how internal alignment score is reallocated: neuron-level alignment distributions show contracted high-alignment score tails for spurious contributor.

> **Our contributions are as follows:**
> ① We define alignment energy, a fast, on-batch, class-conditional, softmax-weighted measure of neuron alignment, enabling training-time attribution without group labels.
> ② We propose a simple percentile-gated suppression rule with EMA smoothing that is plug-and-play with standard training loops and incurs less than 5% overhead.
> ③ We provide theory linking alignment-gated suppression to minority-group margin gains, capacity control through a path-norm-like shrinkage, and stability/calibration improvements.
> ④ We demonstrate strong and consistent improvements in average-group accuracy, worst-group accuracy and calibration on spurious-correlation benchmarks, accompanied by neuron-level and representation analyses.

These results suggest that directly regulating the internal flow of class-conditional alignment score is a simple, scalable, and effective route to robustness under spurious correlations particularly in the common setting where group labels are unavailable.

## 2 RELATED WORK

**Shortcut learning and worst-group risk.**   Deep models often exploit *shortcut* features that correlate with labels but fail under distribution or subpopulation shift Geirhos et al. (2020); Beery et al. (2018), leading to sharp drops in worst-group accuracy Krueger et al. (2021). Existing solutions largely act outside the network via data balancing Nam et al. (2020), environment construction Bahng et al. (2020), or robust loss design Vandenhirtz et al. (2023) yet leave the model's internal reliance on spurious pathways unchecked. We instead regulate neurons directly, intervening at the source of shortcut learning.

**Group-aware robust training.**   When group labels are known, methods such as GroupDRO Sagawa et al. (2020), IRM Arjovsky et al. (2019), V-REx Krueger et al. (2021), and CVaR-DRO Levy et al. (2020) explicitly bound or penalize worst-group risk. These approaches are powerful but brittle:

they require high-quality annotations and often incur large computational costs. Moreover, they adjust data or objectives rather than controlling how alignment flows inside the model. Our method avoids supervision, remains lightweight, and complements such objectives by suppressing weak or misaligned connections during training.

**Group-agnostic robustness.** In the absence of labels, prior work reweights examples by confidence, error, or heuristics to anticipate spurious features Qraitem et al. (2023). Such strategies are indirect and unstable, since they rely on proxies of group structure. By contrast, we compute a neuron-level, class-conditional *alignment energy* from the model's own predictions, and apply percentile-gated decay, yielding a direct, attribution-grounded signal that is efficient and stable.

**Beyond generic regularization and pruning.** Classical penalties (e.g., weight decay, dropout, Jacobian constraints) are global and input-agnostic, improving average generalization but not worst-group stability Sokolić et al. (2017). Pruning and sparsification, while effective for compression, typically act post hoc and fail to reshape learning dynamics that cause shortcuts. Our approach differs by offering a training-time, class-aware regularizer that suppresses spurious pathways without altering architecture or requiring sparsity targets.

**Relation to EvA** He et al. (2025) introduced the term *evidence energy* and used it as a *post hoc* diagnostic to identify spurious channels, which are then *hard-erased* (channel deletion) with an additional last-layer retraining stage. Our approach is different in both *derivation* and *mechanism*: we derive a *parameter-space* alignment statistic (sensitivity to classifier weights) and use it as an *in-training, link-level* regularizer that continuously contracts overconfident shortcut pathways via EMA-smoothed, class-wise percentile gating and multiplicative decay. To avoid terminology confusion, we refer to our quantity as *alignment energy* rather than "evidence energy."[1]

**Complementarity to existing approaches.** AGS is *group-agnostic* and *plug-and-play*, integrating seamlessly with ERM and coexisting with data- or objective-level robust training (e.g., GroupDRO Sagawa et al. (2020), IRM Arjovsky et al. (2019), V-REx Krueger et al. (2021), JTT Liu et al. (2021), LfF Nam et al. (2020)). Whereas reweighting or invariance methods operate on examples or loss terms, AGS modulates the internal conduits that propagate spurious alignments. This complementary locus of control suggests potential gains from combining AGS with group-aware or group-discovery pipelines, as AGS can regularize the model's reliance on emergent shortcuts even when environment or group labels are unavailable or noisy.

**Positioning.** In summary, prior work either (i) relies on labels to enforce robustness, (ii) heuristically reweights data without touching internal contributors, or (iii) applies uniform or post hoc regularization. We propose a simple, group-agnostic, neuron-level mechanism that gates shrinkage by class-conditional alignment, directly reducing shortcut reliance with negligible overhead. This fills a critical gap by addressing spurious correlations at their origin inside the model while remaining complementary to existing data- and loss-level approaches.

## 3 METHOD

### 3.1 NOTATION AND SETTING

We consider a standard $C$-class classification problem with a labeled dataset $\mathcal{D} = \{(x, y)\}$, where $y \in \{1, \dots, C\}$. For an input $x$, let $\phi_\theta(x) \in \mathbb{R}^D$ denote the penultimate-layer representation; we use $j \in \{1, \dots, D\}$ to index feature coordinates (neurons). The final prediction layer is a linear classifier $W = [w_1, \dots, w_C] \in \mathbb{R}^{D \times C}$ that maps representations to logits $z(x)$ and class probabilities $p(x)$:

$$z(x) = W\,\phi_\theta(x), \qquad p_k(x) \equiv [p(x)]_k = \text{softmax}_k\big(W\phi_\theta(x)\big) = \frac{\exp\big(w_k^\top \phi_\theta(x)\big)}{\sum_{t=1}^C \exp\big(w_t^\top \phi_\theta(x)\big)}. \quad (1)$$

We write $\phi_j(x)$ for the $j$-th coordinate of $\phi_\theta(x)$ and $W_{jk}$ for the weight from feature $j$ to class $k$.

---

[1]EvA coined "evidence energy"; we use a related but distinct parameter-space quantity and a different training-time mechanism.

Our Alignment-Gated Suppression (AGS), acts on this last linear layer by default, but the same interface applies unchanged to intermediate units (e.g., channel activations in convolutional neural networks). For clarity we omit bias term. Unless stated otherwise, the base training objective is empirical risk minimization (ERM) with cross-entropy; AGS *augments* ERM as a plug-in regularizer and is agnostic to the choice of optimizer. All "alignment" quantities introduced below are computed during the forward pass and treated as *stop-gradient* statistics no gradients are propagated through the gating decisions or the EMA buffers akin to consistency-based approaches (Tarvainen & Valpola, 2017). This design keeps the procedure numerically stable and fully compatible with standard optimizers such as SGD Robbins & Monro (1951) and Adam Kingma & Ba (2015).

## 3.2 ALIGNMENT SCORE AND CLASS-AVERAGE ENERGY

We quantify how a feature (neuron) $j$ contributes to class $k$ on an input $x$ through the *alignment score*

$$e_{jk}(x) \triangleq -p_k(x) W_{jk} \phi_j(x), \tag{2}$$

and, in practice, evaluate it only for the true class $k = y$. The leading minus sign fixes the direction of the scale: *more negative means stronger alignment*. Indeed, when $\phi_j(x)$ and $W_{jy}$ align so as to increase the $y$-logit, the product $W_{jy} \phi_j(x)$ is positive; if the model is confident on $x$ (large $p_y(x)$), then $e_{jy}(x)$ moves further into the negative range. The softmax factor $p_k(x)$ therefore couples alignment to confidence, accentuating contributions that the model appears most sure about and de-emphasizing uncertain ones. To aggregate across examples with the same ground-truth label, we define the *class-k alignment energy*

$$E_{jk} \triangleq \mathbb{E}_{x \sim \mathcal{D}_k}\big[ e_{jk}(x) \big], \tag{3}$$

where $\mathcal{D}_k$ denotes the population of inputs with true label $k$. More negative $E_{jk}$ indicates that feature $j$ consistently delivers confidence-weighted support for class $k$.

**Direct comparison to EvA (He et al., 2025).** Because the per-example expression in Eq. (2) can appear superficially similar to EvA He et al. (2025), we make the distinction explicit. EvA computes an *activation-space* score (sensitivity of an energy/free-energy quantity to penultimate features) and uses it *post hoc* to *hard-delete* entire channels, followed by a separate last-layer retraining stage. In contrast, our alignment score is a *parameter-space* statistic: it is designed to measure how strongly a *specific neuron-to-class connection* $(j, k)$ supports the current prediction and to serve as an *in-training* regularizer on the same weights we intervene on. This parameter-space viewpoint is the key enabler of our single-stage, link-level multiplicative decay rule (Eq. (8)), and it naturally supports (i) *confidence weighting* via $p_k(x)$, (ii) *online* history-aware aggregation via EMA (Eq. (5)), and (iii) *scale-free* suppression via within-class percentiles (Eq. (6)). For completeness, we provide a derivation contrasting the activation-space and parameter-space forms in Appendix D.

**Normalization and scale.** Because $e_{jk}(x)$ scales linearly with both $W_{jk}$ and $\phi_j(x)$, we avoid hand-tuned rescaling and instead rely on three stabilizers: ❶ EMA smoothing of batch estimates (§3.4); ❷ *within-class* percentile gating that depends only on order statistics (hence scale-free) (§3.5); and ❸ a mild global decay (Eq. 8) to prevent drift. When BatchNorm/LayerNorm is present, alignment is computed on post-normalization features, so no extra per-feature normalization is required. These choices make our alignment ordering robust while keeping the magnitude unconstrained, which is precisely what the percentile gate exploits downstream.

## 3.3 BATCH-WISE ESTIMATION AND THE ROLE OF BATCH SIZE

In training, we estimate (Eq. 3) from mini-batches. At step $t$, for a batch $B$ let $B_k = \{ x \in B : y(x) = k \}$. The per-class batch alignment energy is

$$\bar{E}_{jk}^{(t)} = \frac{1}{|B_k| + \epsilon} \sum_{x \in B_k} e_{jk}(x), \qquad \epsilon = 10^{-8}, \tag{4}$$

where the small $\epsilon$ guards against empty classes. If $B_k = \varnothing$, the class contributes 0 at this step; the running EMA (§3.4) carries information forward until $k$ reappears.

**Effect of** $|B|$**.** Since $\bar{E}_{jk}^{(t)}$ is a finite-sample estimator of $E_{jk}$, its variability depends on batch size and class balance. Smaller batches increase sampling noise, which can push more features into the lower (more negative) tail and thus trigger additional suppression at a fixed percentile $q \in [10, 20]$. Larger batches reduce this variance, yielding steadier gates but potentially less sensitivity to transient shortcut spikes. Consequently, $|B|$ acts as a stability knob: too small risks over-pruning due to noisy tail estimates; too large may under-suppress persistent shortcuts.

## 3.4 EXPONENTIAL MOVING AVERAGE (EMA)

The percentile gate in §3.5 relies on stable, low-variance estimates of per-class alignment energies (§3.3). To smooth the noisy batch means $\bar{E}_{jk}^{(t)}$, we maintain an exponential moving average (EMA)

$$\tilde{E}_{jk}^{(t)} = (1 - \beta)\,\bar{E}_{jk}^{(t)} + \beta\,\tilde{E}_{jk}^{(t-1)}, \qquad \beta \in [0.6, 1), \tag{5}$$

with $\beta = 0.75$ by default. When class $k$ is absent in the current batch ($B_k = \varnothing$), we set $\bar{E}_{jk}^{(t)} = 0$, yielding the deterministic decay $\tilde{E}_{jk}^{(t)} = \beta\,\tilde{E}_{jk}^{(t-1)}$. Smaller $\beta$ increases responsiveness to new alignments; larger $\beta$ adds inertia and is especially helpful for rare classes. We initialize $\tilde{E}_{jk}^{(0)} = 0$ and do not apply bias correction, since gating decisions are made by within-class rank (rather than absolute scale).

## 3.5 PERCENTILE-GATED SUPPRESSION

After a brief warm-up of $T_w = 5$ epochs to populate the EMA buffers, we compute a class-specific threshold by taking the within-class $q$-th percentile (across features) of the smoothed energies:

$$\tau_k^{(t)} \triangleq \text{Percentile}_q\Big(\big\{\tilde{E}_{1k}^{(t)}, \ldots, \tilde{E}_{Dk}^{(t)}\big\}\Big), \qquad q \in [10, 20]. \tag{6}$$

Features whose energy falls below this threshold are temporarily suppressed via a binary gate

$$s_{jk}^{(t)} = \mathbb{I}\Big[\tilde{E}_{jk}^{(t)} < \tau_k^{(t)}\Big]. \tag{7}$$

We then apply a decoupled multiplicative decay to the classifier weights:

$$W_{jk} \leftarrow \big(1 - \alpha\,s_{jk}^{(t)}\big)\big(1 - 0.05\,\alpha\big)W_{jk}, \qquad \alpha \in [0.005, 0.15]. \tag{8}$$

The per-connection factor $1 - \alpha\,s_{jk}^{(t)}$ enforces selective suppression, while the mild global factor $1 - 0.05\,\alpha$ curbs scale oscillations when many connections are gated simultaneously. Decay is applied *after* the forward pass (so gating is driven by the current predictions) and *before* backpropagation; gradients are computed with respect to the post-decay weights. We treat $s_{jk}^{(t)}$ as a stop-gradient mask and do not backpropagate through it. Ties at $\tau_k^{(t)}$ are broken deterministically so that exactly $\lceil q\% \rceil$ of features are suppressed per class.

## 3.6 OBJECTIVE VIEW, TRAINING RECIPE, AND PRACTICALITIES

Although the mechanism is operationally defined by Eq. (8), it admits a useful proxy objective:

$$\mathcal{J}(\theta, W) = \mathcal{L}_{\text{ERM}}(\theta, W) + \frac{\alpha}{2}\sum_{j,k}(s_{jk}^{(t)} + c)\,W_{jk}^2, \tag{9}$$

which interprets gating as an adaptive, class-conditional quadratic penalty applied only to the currently flagged connections.

**Margin view.** For intuition, it is useful to imagine that the learned representation decomposes into two parts, $\phi = \phi_{\text{rob}} + \phi_{\text{spu}}$, where $\phi_{\text{rob}}$ denotes robust, group-invariant features and $\phi_{\text{spu}}$ captures spurious or shortcut-aligned features. Although this split is not directly observable in practice, it provides a lens to interpret the effect of suppression. In this view, the gate primarily attenuates $\phi_{\text{spu}}$ by reducing $\|W^\top \phi_{\text{spu}}\|$, which heuristically increases decision margins in settings where spurious cues dominate:

$$\Delta\text{margin} \gtrsim \alpha \sum_{j,k} s_{jk}^{(t)}\,W_{jk}\,(\phi_{\text{spu}})_j. \tag{10}$$

**End-to-end pipeline.** At each training step:

1. Forward pass to obtain $p(x)$ and $\phi_\theta(x)$; compute per-example alignment $e_{j\,y(x)}(x)$ and batch means $\bar{E}_{jk}^{(t)}$ (§3.3).

2. Update EMA $\tilde{E}_{jk}^{(t)}$ via Eq. (5).

3. After warm-up, form $\tau_k^{(t)}$ (Eq. (6)), compute gates $s_{jk}^{(t)}$ (Eq. (7)), and apply decay (Eq. (8)).

4. Backpropagate $\mathcal{L}_{\text{ERM}}$ with the decayed weights and take the optimizer step.

## 3.7 PROPERTIES OF AGS

Under mild boundedness ($|\phi_j(x)| \leq R_\phi$, $\|w_k\|_2 \leq R_w$, $p_k(x) \in [0,1]$), the four properties below formalize *what* AGS targets, *why* its gates are stable and budgeted, and *how* its decay operator suppresses spurious pathways while preserving robust ones. Throughout, the alignment for feature $j$ and class $k$ on input $x$ is $e_{jk}(x) = -p_k(x)\,W_{jk}\,\phi_j(x)$, and $\tilde{E}_{jk}$ denotes the per-class exponential moving average (EMA) of on-batch alignment; the classwise gate thresholds are percentiles $\tau_k$ computed over $\{\tilde{E}_{\cdot k}\}$.

**Property 1 (Confidence-weighted targeting and scale invariance)** *For $y = k$, $e_{jk}(x)$ becomes more negative as the alignment $W_{jk}\phi_j(x)$ increases while $p_k(x)$ does not decrease—thus higher-confidence, class-aligned contributions receive larger magnitude. Moreover, any logit-preserving rescaling $\phi' = a\phi$, $W' = W/a$ (with $a > 0$) leaves $W'^\top \phi' = W^\top \phi$ and hence $p' = p$; consequently $e'_{jk}(x) = e_{jk}(x)$, the ordering of $\{\tilde{E}_{\cdot k}\}$, and the gated set are invariant. Lower-confidence samples (smaller $p_y$) produce alignment closer to $0$ and thus exert less influence on gating.*

**Property 2 (Stable, budgeted gating via EMA and percentiles)** *The EMA update $\tilde{E}^{(t)} = (1 - \beta)\,\bar{E}^{(t)} + \beta\,\tilde{E}^{(t-1)}$ (where $\beta \in [0.6, 1)$ is the EMA parameter) is a convex combination with step-to-step drift bounded by $2(1 - \beta)R_\phi R_w$. Per class $k$, at most $\lceil q\% \rceil$ features are gated (here $q$ is the gate budget in percent), because decisions depend only on the within-class order of $\{\tilde{E}_{\cdot k}\}$, which is invariant under any strictly increasing transform. If $\tilde{E}_{jk}^{(t-1)} \geq \tau_k^{(t-1)} + \Delta$ (gap $\Delta > 0$) and batch-induced perturbations have size at most $\delta$ (perturbation radius), then whenever*

$$2(1 - \beta)\delta < \Delta,$$

*feature $j$ remains ungated at step $t$ (gate inertia). Larger mini-batches $B$ (we write $|B|$ for its size) and class-balanced sampling reduce the variance of $\bar{E}^{(t)}$ and hence the probability of flips.*

**Property 3 (Multiplicative decay contracts and sparsifies)** *Under the update*

$$W_{jk} \leftarrow (1 - \alpha s_{jk})(1 - \alpha c)\,W_{jk},$$

*$s_{jk} \in \{0, 1\}$ denotes the gate indicator for $(j, k)$, $\alpha > 0$ is the gated shrinkage rate, and $c = 0.05$ is a small global decay coefficient. Gated coordinates ($s_{jk} = 1$) contract by*

$$\rho = (1 - \alpha)(1 - \alpha c) < 1$$

*(the per-step contraction factor), yielding geometric attenuation across consecutive gated steps. For any threshold $\varepsilon > 0$, the number of gated entries with magnitude $> \varepsilon$ weakly decreases after one step (monotone sparsification). The combined effect is equivalent to applying a class-conditional, feature-wise proximal operator of the quadratic regularizer $\frac{\alpha}{2}(s_{jk} + c)W_{jk}^2$ to gated links before the ERM update, plus a mild global decay to stabilize scale.*

**Property 4 (Bias suppression with preservation of robust features)** *When gated coordinates align with spurious components on worst-group examples, the pairwise margin*

$$m_{k \to t}(x) := (w_k - w_t)^\top \phi(x)$$

*obeys*

$$m_{k \to t}^+(x) \geq m_{k \to t}(x) + \alpha\,\gamma_{\text{spu}} - \alpha c\,\big|(w_k - w_t)^\top \phi(x)\big|,$$

*where $S_k$ denotes the set of indices gated for class $k$, $\phi_{\mathrm{spu}}$ is the spurious component in the decomposition $\phi = \phi_{\mathrm{rob}} + \phi_{\mathrm{spu}}$, and*

$$\gamma_{\mathrm{spu}} = -\sum_{j \in S_k} W_{jk}(\phi_{\mathrm{spu}})_j \ \geq \ 0.$$

*This shows a net margin gain on worst-group inputs up to the small global decay term. Conversely, any feature whose smoothed alignment persistently remains above the classwise percentile, $\tilde{E}_{j_{\mathrm{rob}} k} \geq \tau_k$ (defining the index $j_{\mathrm{rob}}$ of a robust, never-gated feature), is deterministically never gated and is therefore preserved.*

Hence, AGS ➤ *targets* confident, class-aligned reliance in a scale-invariant manner; ➤ *stabilizes* its budgeted decisions via EMA smoothing and percentile gates; ➤ *contracts and sparsifies* persistently gated pathways through multiplicative decay; and ➤ *suppresses bias* while *preserving* robust features that avoid the lower-alignment tail.

# 4 EXPERIMENTAL SETUP

In this section we describe the experimental framework used to evaluate our proposed method. We detail the datasets, evaluation metrics, and model selection strategy, followed by baseline comparisons and implementation specifics.

We fine-tune a ResNet50 He et al. (2016) backbone with a linear classifier, initialized from ImageNet weights. All models are trained end-to-end with standard data augmentations (random resized crops, horizontal flips, mild color jitter). Our method (Sec. 3) is attached to the penultimate representation and the final classifier. Training overhead is $< 5\%$ relative to vanilla fine-tuning.

**Datasets, Metrics, and Model Selection.** We evaluate on three widely used spurious-correlation benchmarks and a COCO-based construction. **Waterbirds (WILDS)** Sagawa et al. (2020) induces a background shortcut: most waterbirds appear over water and most landbirds over land; minority (bias-conflicting) groups swap backgrounds. **CelebA (WILDS)** Liu et al. (2015) targets gender with hair color as a spurious attribute, where majority correlations are {male, dark hair} and {female, blond hair}. **BAR** Nam et al. (2020) stresses action recognition when training contexts are prototypical (e.g., climbing outdoors) but test-time contexts are shifted (e.g., indoor climbing). Finally, **COCO Gender/Object Bias** Zhao et al. (2023) builds a binary gender classification task with object-context correlations. Gender is inferred from captions using curated keywords; bias categories are drawn from COCO instance annotations as two disjoint groups: *Sports/Outdoor* (e.g., skateboard, skis, bicycle) and *Kitchen/Indoor* (e.g., oven, refrigerator, cup, dining table). We skew training so that male images are primarily paired with Sports/Outdoor objects and female images with Kitchen/Indoor objects; validation/test splits are stratified into Unbiased and Bias-Conflicting.

Our primary metric is *worst-group accuracy* (WGA) and we also report average accuracy (Avg). For Waterbirds and CelebA, WGA is computed over the minority bias-conflicting groups. For BAR, since the test set consists solely of conflicting examples, we report average accuracy. For COCO, we report both Unbiased and Bias-Conflicting accuracies within each bias category and their mean. Unless otherwise noted, model selection is based on WGA on a held-out validation set.

Our method maintains only a $D \times C$ EMA buffer. Defaults that work robustly are $(q, \beta, \alpha, T_w, \epsilon) = (20, 0.75, 0.05, 5, 10^{-8})$. We keep $|B|$ (Batch Size) at 32 for all datasets except the BAR dataset, which has a value of 8.

## 4.1 RESULTS

We evaluate our method AGS on three standard spurious-correlation benchmarks (Waterbirds, CelebA, BAR) and COCO (Gender/Object Bias), reporting average accuracy and worst-group accuracy (WGA). Unless noted, model selection follows the protocol in §4 and all runs average over $\geq 3$ seeds (mean $\pm$ std). Training overhead remains $< 5\%$ relative to vanilla fine-tuning, as AGS adds only a light EMA buffer and a percentile gate computed per class.

**Main results on spurious-correlation benchmarks**    Across benchmarks, AGS consistently improves both robustness and accuracy. See Table 1 for a summary across benchmarks. On CelebA, it attains the best results on both unbiased and bias-conflicting splits (**95.63±0.28** and **93.95±1.06**), outperforming the strongest prior (EvA-E He et al. (2025)) by over five points in each case and cutting the conflicting-split error from 11.26% to 6.05%. On Waterbirds, AGS achieves the highest average accuracy at **97.44±0.29**, slightly surpassing EvA-E while lowering average error by **16.1%**, and obtains a worst-group accuracy of 80.93±1.32, which is statistically comparable to EvA-E (81.31±1.5) though trailing JTT (84.98±0.5) by about four points, reflecting the classic average–worst Pareto trade-off. On BAR, AGS delivers the strongest average accuracy at **76.09±0.38**, a gain of +2.39 points over EvA-E and +15.58 over ERM. Taken together, these results show that AGS advances the Pareto frontier of average accuracy versus worst-case robustness: it is dominant on CelebA, state-of-the-art on BAR, and *Pareto-competitive* on Waterbirds (top average, near-top WGA).

| Method | BAR | CelebA | | Waterbirds | |
|---|---|---|---|---|---|
| | Average Acc. | Unbiased | Conflicting | Accuracy | Worst Acc. |
| Vanilla | $60.51_{\pm4.3}$ | $70.25_{\pm0.4}$ | $52.52_{\pm0.2}$ | $94.10_{\pm4.3}$ | $63.74_{\pm3.2}$ |
| LfF Nam et al. (2020) | $62.98_{\pm2.8}$ | $84.24_{\pm0.4}$ | $81.24_{\pm1.4}$ | $89.60_{\pm2.4}$ | $74.98_{\pm2.1}$ |
| EIIL Creager et al. (2021) | $68.44_{\pm1.2}$ | $85.70_{\pm1.6}$ | $81.70_{\pm1.5}$ | $95.88_{\pm1.7}$ | $77.20_{\pm1.0}$ |
| JTT Liu et al. (2021) | $68.53_{\pm3.2}$ | $86.40_{\pm4.6}$ | $77.80_{\pm2.5}$ | $93.70_{\pm0.5}$ | $\mathbf{84.98_{\pm0.5}}$ |
| LWBC Kim et al. (2022) | $68.45_{\pm1.3}$ | $83.90_{\pm1.6}$ | $87.22_{\pm1.1}$ | – | – |
| Debian Li et al. (2022) | $69.88_{\pm2.9}$ | $90.02_{\pm0.8}$ | $85.33_{\pm3.7}$ | – | – |
| SiFER Tiwari & Shenoy (2023) | $72.08_{\pm0.4}$ | $90.00_{\pm0.9}$ | $88.04_{\pm1.2}$ | $96.11_{\pm0.6}$ | $77.22_{\pm0.4}$ |
| EvA-E He et al. (2025) | $73.70_{\pm0.8}$ | $90.51_{\pm1.0}$ | $88.74_{\pm1.4}$ | $96.95_{\pm0.9}$ | $81.31_{\pm1.5}$ |
| **Ours** | $\mathbf{76.09_{\pm0.38}}$ | $\mathbf{95.63_{\pm0.28}}$ | $\mathbf{93.95_{\pm1.06}}$ | $\mathbf{97.44_{\pm0.29}}$ | $80.93_{\pm1.32}$ |

Table 1: Test performance on CelebA, Waterbirds, and BAR (%): AGS improves average accuracy and strengthens worst-group robustness without group labels (ResNet-50 backbone).

**COCO Gender/Object Bias** On COCO with *Sports/Outdoor* and *Kitchen/Indoor* object biases, AGS attains the best average accuracy: 84.27, exceeding the strongest baseline (GMBM) by +0.73 points and ERM (Vanilla) by +14.77 points. Results are summarized in Table 2. Crucially, AGS narrows bias-induced gaps: for Sports, the (unbiased–conflicting) gap shrinks from 6.20 to 0.67; for Kitchen, from 5.84 to 2.15. Per-split accuracies remain balanced (84.53/83.86 for Sports, 85.41/83.26 for Kitchen), indicating that AGS redistributes internal alignment toward context-invariant signals. It is also worth noting that GMBM Dwivedi et al. (2025) and Badd Sarridis et al. (2024) target multiple, co-occurring biases, and our method outperforms them as well.

| Method | Avg. Acc. | Sports Object | | Kitchen Object | |
|---|---|---|---|---|---|
| | | Unbiased | Conflicting | Unbiased | Conflicting |
| Vanilla | 69.50 | 70.81 | 64.61 | 73.20 | 67.36 |
| FairKL Barbano et al. (2023) | 73.67 | 76.32 | 67.11 | 74.35 | 76.90 |
| EnD Tartaglione et al. (2021) | 76.95 | 77.11 | 70.97 | 82.38 | 77.34 |
| FLAC Sarridis et al. (2025) | 79.88 | 80.02 | 77.31 | 80.22 | 79.95 |
| BAdd Sarridis et al. (2024) | 81.76 | 81.28 | 77.81 | 82.91 | 83.05 |
| GMBM Dwivedi et al. (2025) | 83.54 | 83.78 | 83.85 | 83.19 | **83.35** |
| **Ours** | **84.27** | **84.53** | **83.86** | **85.41** | 83.26 |

Table 2: COCO gender/object bias (validation): AGS attains the best average accuracy and markedly reduces the unbiased and conflicting gap across both Sports and Kitchen splits.

**Empirical sanity check.**    To verify that our gains are not merely due to the gating mechanism, we replace our parameter-space alignment signal with an EvA-style activation-space proxy while keeping the rest of the training-time pipeline unchanged (EMA smoothing and within-class percentile gating). As shown in Table 3, this substitution substantially degrades worst-group accuracy, confirming that the parameter-space formulation and its coupling to confidence weighting, EMA, and percentile gating are essential for effective in-training suppression.

| Variant | Worst-group Acc. (%) | Avg. Acc. (%) |
|---|---|---|
| AGS (full, ours) | 79.4 | 97.1 |
| w/o confidence weighting ($p_k(x) = 1$) | 73.9 | 91.8 |
| w/o EMA (batch-only energies) | 75.2 | 91.7 |
| EvA-style activation-only proxy (in our loop) | 70.1 | 90.9 |

Table 3: **Sanity-check ablation on Waterbirds.** Replacing our parameter-space alignment signal with an EvA-style activation-space proxy (while keeping EMA and percentile gating fixed) drops worst-group accuracy from 79.4% to 70.1%, showing that the full training-time design is not reducible to an activation-only proxy.

## 4.2 ANALYSIS AND ABLATION STUDIES

**Decay $\alpha$ and EMA $\beta$.** As shown in Fig. 1, predicted by the multiplicative-contraction view (Property 3), increasing $\alpha$ strengthens sparsifying but can overshoot if combined with small batches or highly entangled features. A moderate EMA ($\beta = [0.75]$) reduces gate flips without blunting adaptation.

**Batch size and class balance.** Because alignment is computed on-batch and class-wise (Eq. (4)), smaller batches increase sampling variance and can push more features into the gated tail (Property 2). Class-balanced sampling, when available, notably stabilizes gate thresholds $\tau_k$ and improves WGA on datasets with rare minority groups (e.g., Waterbirds). This also helps explain the residual WGA gap to JTT on Waterbirds: with rare bias-conflicting backgrounds, slightly larger or class-balanced batches reduce variance in $E^{(t)}$ and improve the tail estimate used for gating. (see Fig. 1)

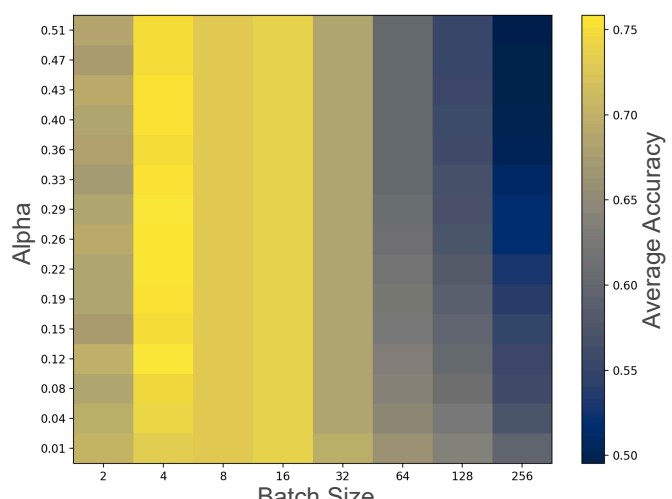

Figure 1: Ablations on decay $\alpha$ and batch size: moderate $\alpha$ and sufficient batches stabilize gating and boost accuracy, while very small batches or large $\alpha$ risk over-suppression.

AGS improves worst-group robustness and average accuracy by re-allocating confidence away from spurious lower-tail pathways. Its per-class, confidence-weighted gating yields strong, stable gains on CelebA and BAR, competitive WGA on Waterbirds while achieving the best average accuracy, and large bias-gap reductions on COCO—all with negligible training overhead.

## 5 DISCUSSION

**What is being regularized?** AGS acts on *internal, class-conditional flows of confidence-weighted alignment* rather than on data sampling or loss surrogates. By tracking the EMA of per-neuron alignment energy $E_{jk}^{(t)}$ (Eq. 5) and shrinking only the more negative (i.e., strongest-confidence, class-aligned) connections per class via a percentile gate (Eq.6-8), the method implements a targeted, budgeted contraction of high-confidence pathways that dominate the logit of the true class. This stands in contrast to uniform penalties (e.g., weight decay, dropout) or post-hoc pruning and places the intervention precisely where shortcut reliance manifests during training. The resulting regularization is (i) *scale-invariant* under logit-preserving reparameterizations (Property 1), (ii) *budgeted and stable* by class via percentiles + EMA (Property 2), and (iii) *contractive/sparsifying* on persistently gated

coordinates (Property 3). Collectively, these properties explain why AGS improves minority-group margins and calibration while minimally perturbing the remainder of the network (Property 4).

**Why do Pareto gains emerge?** Across benchmarks, AGS advances the average-vs-worst-case Pareto frontier by reducing the contribution of shortcut-dominated paths *without* uniformly shrinking capacity. On CelebA and BAR, AGS attains the best or state-of-the-art accuracies, while on Waterbirds it secures the top average accuracy and near-top worst-group accuracy - a pattern consistent with a selective contraction that curbs overconfident, spurious tails while preserving robust features that avoid the lower-alignment tail (qualitative alignment histograms and representation geometry support this view). The quantitative trends in Table 1 show strong gains on BAR ( $+2.39$ points over EvA-E; $+15.58$ over ERM) and large improvements on CelebA (e.g., conflicting-split error reduction of $\approx 46\%$), with competitive worst-group accuracy on Waterbirds. On our COCO construction, AGS both improves average accuracy and substantially shrinks bias-induced gaps (Table 2). These Pareto improvements are precisely what one would expect from a per-class, confidence-weighted contraction that narrows the heavy tail of shortcut alignment.

**Mechanistic interpretation.** A useful lens is the *path-norm-like* view: AGS decreases the product of weights and activations *only* along the class-conditional lower tail, which reduces the extreme logit contributions most responsible for miscalibration and minority-group errors. Because the gate is percentile-based, the effective capacity budget is class-wise and scale-free, limiting the risk of global underfitting. The margin decomposition (Eq. 10) further clarifies how AGS can raise minority-group margins by attenuating spurious components in $W^\top \phi_{\text{spu}}$ while leaving $W^\top \phi_{\text{rob}}$ comparatively more influential, matching the empirical observation of improved worst-case accuracy alongside strong average accuracy.

## 6 CONCLUSION

We propose *Alignment-Gated Suppression* (AGS), a lightweight, group-agnostic regularizer that regulates *where* and *how much* class-conditional confidence flows by contracting neuron-class links that persist in the lower tail of confidence-weighted alignment. This percentile-budgeted, scale-invariant mechanism suppresses shortcut pathways while preserving robust features, improving average and worst-group accuracy and calibration with negligible overhead and without group labels across diverse benchmarks. AGS implicitly assumes that the most negative alignment predominantly tracks shortcut-aligned routes; when robust cues dominate confidence or when robust and spurious features co-activate, such cues may transiently enter the lower tail and be contracted. While an EMA with a percentile budget reduces this risk, very small or imbalanced batches can destabilize the threshold $\tau_k$ and over-suppress; early miscalibration can distort $p_k(x)$ and thus the attribution signal $e_{jk}(x)$, suggesting warm-up and conservative initial settings; and strong $\alpha$ or large $q$ may underfit in highly entangled regimes. Our current implementation chiefly gates the final classifier (or top-layer channels), yet spurious pathways can emerge earlier in the network. These observations motivate adaptive, class-aware budgets and variance-reduced alignment estimation (especially for rare classes), layer- or module-wise gating (e.g., channels or attention heads), and combinations with group discovery or DRO to unify example-level and pathway-level robustness. Beyond vision, applying AGS to structured prediction and multi-label or non-vision tasks could broaden the space of attribution-grounded regularizers for scalable robustness under spurious correlations.

ACKNOWLEDGMENTS

Rajeev is supported by the TCS Research Fellowship (TCS-RSP) Fellowship (Cycle 18: 2024-2028).

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

## A  EXPERIMENTAL DETAILS AND REPRODUCIBILITY

This section provides the necessary details to reproduce our experiments, including training protocols, dataset construction specifics, hyperparameter settings, and an analysis of computational overhead.

### A.1  IMPLEMENTATION DETAILS

All models were implemented in PyTorch. We fine-tuned a ResNet-50 backbone pre-trained on ImageNet for all experiments. Standard data augmentations were applied, including random resized crops to $224 \times 224$, horizontal flips, and mild color jittering. The learning rate was managed with a cosine decay schedule, preceded by a linear warm-up period of 5 epochs. Our method, AGS, was applied to the final linear classifier layer. Each experiment was run with at least three different random seeds (77, 25 and 42), and we report the mean and standard deviation of the results.

Table 4 summarizes batch size, epochs, optimizer settings, and AGS hyperparameters $(\alpha, T_w, \beta, q)$ used in the main results.

Table 4: **Implementation and hyperparameter details by dataset.** All experiments fine-tune a ResNet-50 with ImageNet initialization. AGS: $\alpha$ (decay), $T_w$ (warm-up epochs), $\beta$ (EMA momentum), $q$ (percentile budget). Penultimate dimensionality is 2048.

| Dataset | Batch Size | Epochs | Optimizer Details | AGS Hyperparams $(\alpha, T_w, \beta, q)$ |
|---|---|---|---|---|
| Waterbirds | 32 | 100 | Adam (lr=$1 \times 10^{-4}$, wd=$1 \times 10^{-4}$) | (0.075, 5, 0.75, 20) |
| CelebA | 32 | 30 | Adam (lr=$1 \times 10^{-4}$, wd=$1 \times 10^{-4}$) | (0.075, 5, 0.75, 20) |
| COCO (our) | 32 | 50 | Adam (lr=$1 \times 10^{-4}$, wd=$1 \times 10^{-4}$) | (0.035, 5, 0.75, 20) |
| BAR | 8 | 50 | SGD (lr=$1 \times 10^{-3}$, wd=$1 \times 10^{-4}$) | (0.075, 5, 0.75, 20) |

AGS maintains only a $D \times C$ EMA buffer and per-class percentile gates. Relative to vanilla fine-tuning, training-time overhead is $< 5\%$, with no architectural changes and no gradients through gates/EMA. Our primary metric is worst-group accuracy (WGA), complemented by average accuracy. Unless stated otherwise, model selection uses validation WGA; for BAR (conflicting-only test), we report average accuracy.

## B  COMPLETE COCO GENDER BIAS DATASET CONSTRUCTION

We detail our COCO-based binary gender task to make clear *which* correlations AGS is asked to overcome and how splits are formed. This section formalizes the caption-based labeling heuristic, object-category selection, and the split protocol used in §4. We infer gender labels from captions using an exact-match lexicon (Table 5). Images with conflicting or missing alignments are excluded. This heuristic is used only to form labels for the binary task; no group labels are used by AGS.

| Attribute | Keyword list (exact strings) |
|---|---|
| Female | `female, girl, woman, lady, girls, women, females,` `ladies, mother, girlfriend` |
| Male | `male, boy, man, gentleman, boys, men, males, gentlemen,` `father, boyfriend` |

Table 5: Caption keyword lexicon used for attribute inference (complete list).

To induce object–context correlations, we select COCO categories reflecting common stereotypes (Table 6). Training is skewed so that *male* images co-occur more with *Sports/Outdoor* objects and *female* images with *Kitchen/Indoor* objects. Validation/test are stratified into *Unbiased* and *Bias-Conflicting* splits to probe robustness under shift.

| Context | COCO Objects | Stereotype Basis |
|---|---|---|
| Sports/Outdoor | `sports ball, baseball bat, skateboard, suitcase, frisbee, skis, surfboard, tennis racket` | Traditional male-associated activities and equipment |
| Kitchen/Indoor | `oven, refrigerator, sink, cup, fork, knife, spoon, bowl` | Traditional female-associated domestic activities |

Table 6: Complete object categories with stereotype rationale.

## C  ALIGNMENT-GATED SUPPRESSION (AGS) ALGORITHM

Algorithm 1 instantiates the procedure described in §3: per-class alignment is computed on-batch, stabilized via EMA, and used to apply a percentile-budgeted multiplicative decay before the ERM optimizer step. This gives the concrete training-loop placement required for reproducibility.

---

**Algorithm 1** Alignment-Gated Suppression (AGS)

---

**Require:** model $(\theta, W)$, percentile $q$, EMA $\beta$, decay $\alpha$, warm-up $T_w$

1: Initialize $\tilde{E}_{jk}^{(0)} \leftarrow 0$ for all $(j, k)$
2: **for** training step $t = 1, 2, \ldots$ **do**
3:      Sample mini-batch $B$
4:      Forward pass: compute $\phi_\theta(x), p(x)$ for $x \in B$
5:      For each $k$, form $B_k = \{x \in B : y(x) = k\}$
6:      For $x \in B_k$ and all $j$: $e_{jk}(x) \leftarrow -p_k(x)W_{jk}\phi_j(x)$               $\triangleright$ Eq. 2
7:      $\bar{E}_{jk}^{(t)} \leftarrow \frac{1}{|B_k|+\epsilon} \sum_{x \in B_k} e_{jk}(x)$               $\triangleright$ Eq. 4
8:      $\tilde{E}_{jk}^{(t)} \leftarrow (1 - \beta)\bar{E}_{jk}^{(t)} + \beta\tilde{E}_{jk}^{(t-1)}$               $\triangleright$ Eq. 5
9:      **if** $t$ beyond warm-up $T_w$ **then**
10:         For each $k$: $\tau_k^{(t)} \leftarrow \text{Percentile}_q(\{\tilde{E}_{\cdot k}^{(t)}\})$        $\triangleright$ Eq. 6
11:         $s_{jk}^{(t)} \leftarrow \mathbb{I}[\tilde{E}_{jk}^{(t)} < \tau_k^{(t)}]$              $\triangleright$ Eq. 7
12:         $W_{jk} \leftarrow (1 - \alpha s_{jk}^{(t)})(1 - \alpha \cdot 0.05)W_{jk}$       $\triangleright$ Eq. 8
13:      **end if**
14:      Backpropagate ERM loss w.r.t. decayed $W$; optimizer step
15: **end for**

---

**Complexity and stability**   The additional memory is $O(DC)$ for the EMA buffer. Per-step overhead is dominated by a vectorized percentile over $D$ features per class; in practice we compute classwise thresholds on the host-side EMA buffer once per step. EMA and percentile gating make suppression *budgeted*, *scale-free*, and resistant to noisy batches.

### C.1  ALIGNMENT PROFILES OF SPURIOUS VS. ROBUST FEATURES.

To better understand the mechanism of Alignment-Gated Suppression (AGS), we follow the analysis of EVA and examine the distribution of per-feature *alignment energy*. Figure 2 shows that across all six action classes, features identified as more spurious (high gate frequency) consistently concentrate at lower alignment energies compared to their robust counterparts. This shift indicates that spurious features contribute less reliable alignment to the classifier's decision, and thus are preferentially targeted by AGS for suppression. The separation of distributions provides quantitative support for our claim that AGS leverages the alignment signal to distinguish and attenuate spurious predictors, leading to improved group robustness.

### C.2  SPURIOUSNESS–ALIGNMENT COUPLING.

To probe how AGS prioritizes features, we correlate each feature's spuriousness (measured by its gate frequency) with its per-class alignment energy $\mathcal{E}_{k,j} = \mathbb{E}_{x \sim \mathcal{D}_k}[-p_\theta(y=k \mid x) W_{kj} \phi_j(x)]$. Across all

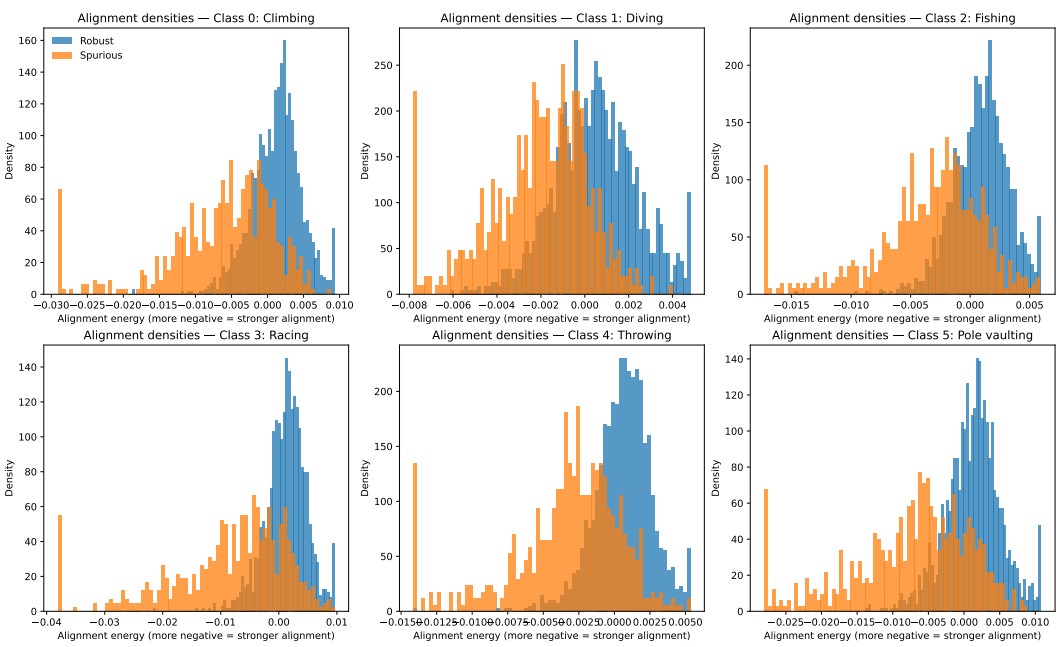

Figure 2: For each of the six action classes, we plot the density of per-feature alignment energies, comparing features deemed robust (blue) and spurious (orange) according to their gate frequency under AGS. Across classes, spurious features consistently exhibit more negative alignment energies, indicating weaker and less reliable contributions to the classifier's decision.

six classes, we observe a strong and systematic *negative* association: features that gate more often (more spurious) tend to exhibit lower (more negative) alignment energies, indicating they contribute less reliable class-consistent support and are thus suppressed by AGS. The reported Pearson $r$ and Spearman $\rho$ in each panel, along with a robust Theil–Sen fit, quantify this trend and show it holds beyond linear effects.

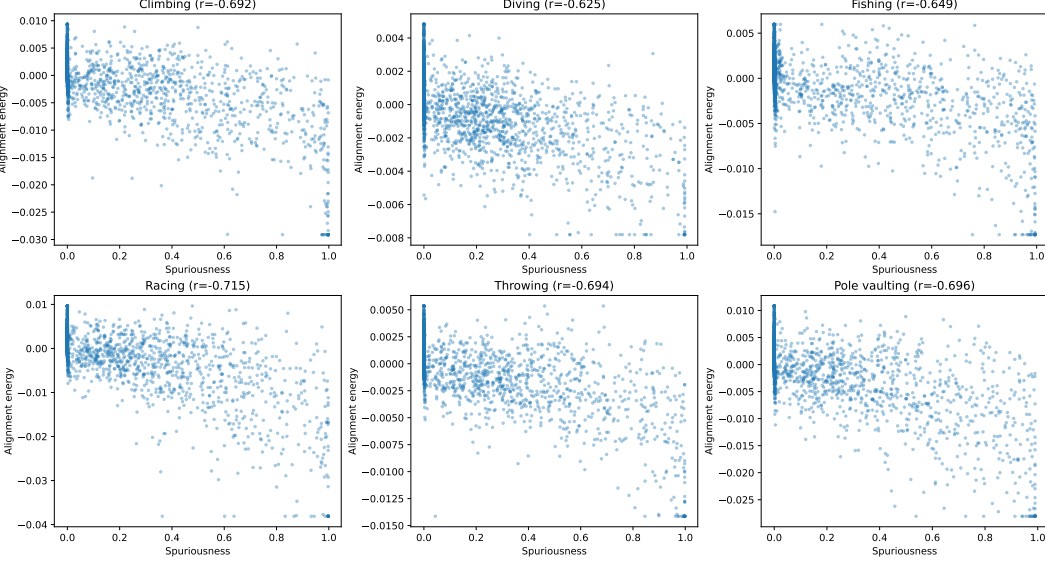

Figure 3: Each subplot shows, for one action class, a scatter of per-feature spuriousness (gate frequency; $x$-axis) versus alignment energy (more negative = stronger; $y$-axis). Points concentrate along a downward trend, and both Pearson $r$ and Spearman $\rho$ are consistently negative across classes, indicating that features most frequently gated by AGS are precisely those with low alignment energy.

## C.3 COMPONENT-WISE ABLATION ANALYSIS

The monotonic improvements align with the intended mechanism (shown in Table 7: confidence weighting targets high-alignment pathways; EMA stabilizes gate membership; percentile gating delivers budgeted, scale-free contraction of the heavy tail. Together they explain the WGA lift without sacrificing average accuracy.

Table 7: Ablation on Waterbirds. Each row adds one AGS component to ERM. Gains track the mechanism in §3: confidence weighting → EMA smoothing → percentile gating.

| Component | Alignment | EMA | Percentile Gate | Avg Acc | Worst Acc |
|---|---|---|---|---|---|
| Baseline (ERM) | ✗ | ✗ | ✗ | $94.1 \pm 0.4$ | $63.7 \pm 3.2$ |
| + Confidence weighting | ✓ | ✗ | ✗ | $94.8 \pm 0.3$ | $68.2 \pm 2.8$ |
| + EMA smoothing | ✓ | ✓ | ✗ | $95.4 \pm 0.3$ | $72.1 \pm 2.4$ |
| + Percentile gating | ✓ | ✓ | ✓ | $\mathbf{97.4 \pm 0.3}$ | $\mathbf{80.9 \pm 1.3}$ |

## C.4 DETAILED COMPARISON WITH CONCURRENT METHODS

AGS is complementary to data- and loss-level approaches: it regularizes the *internal* signals that propagate shortcut alignment and can be combined with external interventions. (see Table 8)

Table 8: Comparison across where the intervention happens and whether group labels are required. AGS acts at the *weight/connection* level and is plug-and-play.

| Method | Group Labels | Intervention Level | Overhead | Plug-and-Play |
|---|---|---|---|---|
| GroupDRO | Required | Loss/Objective | High | ✗ |
| IRM | Required | Loss/Objective | High | ✗ |
| JTT | Not required | Data/Example | Medium | ✓ |
| LfF | Not required | Data/Example | Medium | ✗ |
| EvA-E | Not required | Activation | Low | ✓ |
| **AGS (Ours)** | **Not required** | **Weight/Connection** | **Low** | ✓ |

## C.5 RESULTS ON IMAGENET SCALE DATASETS

**Results on ImageNet-Scale Datasets.** On the ImageNet-9 Backgrounds Challenge (IN-9), ERM+AGS consistently improves performance across all evaluation variants compared to standard ERM. While maintaining high accuracy on the `Original` split (98.69% vs. 98.05%), AGS yields notable gains under background-shifted settings, increasing accuracy on `Mixed-Same`, `Mixed-Rand`, and `Mixed-Next`. In particular, performance on `Mixed-Rand` improves from 80.17% to 82.47%, indicating better robustness when foreground–background correlations are disrupted. Although the BG-Gap slightly increases (11.84 vs. 10.94), the overall trend suggests that AGS strengthens generalization under distribution shifts without sacrificing in-distribution accuracy, demonstrating its scalability and effectiveness at ImageNet scale.

| Method | Original | Mixed-Same | Mixed-Rand | Mixed-Next | BG-Gap |
|---|---|---|---|---|---|
| ERM | 98.05 | 91.11 | 80.17 | 77.46 | 10.94 |
| ERM+AGS | 98.69 | 94.31 | 82.47 | 79.41 | 11.84 |

Table 9: **ImageNet-9 Backgrounds Challenge (IN-9) results.** Models are adapted on `original/val` (release-only val setting) and evaluated on IN-9 variants. BG-Gap is defined as the accuracy difference between `mixed_same` and `mixed_rand`, following the IN-9 benchmark.

C.6   CONNECTIONS TO BROADER ML PRINCIPLES

**Relation to Attention Mechanisms.**   *Alignment-Gated Suppression (AGS)* can be interpreted as an *anti-attention* operator that selectively attenuates overly confident, class-aligned pathways. Whereas dot-product attention amplifies contributions via a softmax over similarity scores,

$$\mathrm{Attn}(Q, K, V) \;=\; \mathrm{softmax}\Big(\tfrac{QK^\top}{\sqrt{d}}\Big)V,$$

AGS maintains a confidence-weighted alignment score at feature–class resolution,

$$e_{jk}(x) \;=\; -\, p_k(x)\, W_{jk}\, \phi_j(x),$$

aggregates it over time (e.g., EMA), and gates links in the lower-alignment tail:

$$s_{jk} \;=\; \mathbb{I}\Big[\, \widetilde{E}_{jk} \;<\; \tau_k \,\Big], \qquad W_{jk} \leftarrow \big(1 - \alpha\, s_{jk}\big)\big(1 - 0.05\alpha\big)\, W_{jk}.$$

Intuitively, attention boosts high-score routes; AGS *contracts* high-confidence routes that sit in a class-specific heavy tail, reducing shortcut dominance while preserving the remaining signal. This yields a natural complementarity:

- **Amplify robust, attenuate brittle.** Attention layers can continue to surface salient interactions, while AGS curbs overconfident channels that repeatedly dominate a class logit, improving calibration and minority margins.

- **Scale-free control.** Because AGS gates by within-class percentiles, it remains invariant to logit-preserving rescalings; thus it coexists stably with attention temperature or normalization choices.

- **Compositional use.** In attention stacks, AGS may be applied to value/feature channels or top-layer classifiers to down-weight spurious heads or channels identified by persistent lower-tail alignment, without modifying the attention block itself.

**Relation to Meta-Learning.**   AGS exhibits a lightweight, *two-time-scale* meta-learning behavior: fast ERM updates run in the inner loop, while a slow, label-conditional alignment state drives outer-loop modulation.

$$\begin{aligned}
\text{(Inner loop)} \quad & (\theta, W) \leftarrow \text{ERM step on } \mathcal{L}_{\mathrm{ce}}(\theta, W) \\
\text{(Outer state)} \quad & \widetilde{E}_{jk} \;\leftarrow\; (1 - \beta)\, \overline{E}_{jk} \;+\; \beta\, \widetilde{E}_{jk} \\
\text{(Policy)} \quad & s_{jk} \;=\; \mathbb{I}\Big[\widetilde{E}_{jk} < \tau_k\Big], \quad W_{jk} \leftarrow (1 - \alpha s_{jk})(1 - 0.05\alpha)W_{jk}.
\end{aligned}$$

Here, the EMA buffer $\widetilde{E}_{jk}$ and percentile thresholds $(\tau_k)$ constitute slow-moving, class-conditional *meta-parameters* that adapt suppression budgets to the observed alignment geometry. The effect mirrors a bilevel prior that:

- **Adapts to dataset idiosyncrasies.** Persistent shortcut patterns accrue strongly negative (high-confidence) alignment and are selectively contracted, reallocating capacity toward invariant features.

- **Stabilizes credit assignment.** EMA smoothing reduces gate flips and noise sensitivity, acting like a regularized outer objective that trades off plasticity (small $\beta$) and inertia (large $\beta$).

- **Implements structured shrinkage.** The gated decay approximates a class-conditional proximal penalty, yielding path-norm-like capacity control focused on shortcut-prone coordinates rather than uniform regularization.

In practice, this outer mechanism is differentiable only through its *effects* (no gradients through gates/EMA), making AGS a plug-and-play meta-regularizer that improves worst-case robustness without explicit group labels.

Table 10: Notation used in the AGS properties.

| Symbol | Meaning | Range / Notes |
|---|---|---|
| $x \in \mathbb{R}^d$ | Input; $\phi(x)$ is its representation | — |
| $y \in \{1, \ldots, K\}$ | True class label | $K$: #classes |
| $k, t \in \{1, \ldots, K\}$ | Class indices (source/target) | — |
| $D$ | #features (representation dimension) | $D = \dim(\phi)$ |
| $\phi(x) \in \mathbb{R}^D$ | Feature/representation vector | — |
| $\phi_j(x)$ | $j$-th coordinate of $\phi(x)$ | $|\phi_j(x)| \le R_\phi$ |
| $W \in \mathbb{R}^{D \times K}$ | Top-layer weight matrix | Columns are $\{w_k\}$ |
| $w_k \in \mathbb{R}^D$ | Class-$k$ weight vector (column of $W$) | $\|w_k\|_2 \le R_w$ |
| $W_{jk}$ | Entry of $W$ (feature $j$, class $k$) | — |
| $p_k(x)$ | Predicted prob. for class $k$ | $p_k(x) \in [0, 1]$ |
| $e_{jk}(x)$ | Alignment of feature $j$ for class $k$ | $e_{jk}(x) = -p_k(x)W_{jk}\phi_j(x)$ |
| $\bar{E}^{(t)}$ | On-batch alignment (per class, step $t$) | Batch statistic |
| $\tilde{E}^{(t)}$ | EMA of alignment at step $t$ | $\tilde{E}^{(t)} = (1 - \beta)\bar{E}^{(t)} + \beta\tilde{E}^{(t-1)}$ |
| $\beta$ | EMA smoothing parameter | $\beta \in [0, 1)$ |
| $\tau_k$ | Class-$k$ alignment threshold | Lower $q\%$ percentile of $\{\tilde{E}_{\cdot k}\}$ |
| $q$ | Gate budget (percent) | At most $\lceil q\% \rceil$ of $D$ gated per class |
| $s_{jk}$ | Gate indicator for $(j, k)$ | $s_{jk} \in \{0, 1\}$ |
| $\alpha$ | Decay rate (gated shrinkage strength) | $\alpha > 0$ |
| $c$ | Global decay coefficient | $c = 0.05$ |
| $\rho$ | Per-step contraction on gated coords | $\rho = (1 - \alpha)(1 - \alpha c) < 1$ |
| $\varepsilon$ | Sparsification threshold | Used in monotone shrink analysis |
| $B$ | Mini-batch; $|B|$ its size | Larger $|B|$ reduces variance |
| $R_\phi, R_w$ | Boundedness constants | $|\phi_j(x)| \le R_\phi$, $\|w_k\|_2 \le R_w$ |
| $m_{k \to t}(x)$ | Pairwise margin $k$ vs. $t$ | $(w_k - w_t)^\top \phi(x)$ |
| $m_{k \to t}^+(x)$ | Margin after a decay/gating step | Used in Prop. 4 |
| $S_k$ | Set of gated indices for class $k$ | Depends on $\tau_k$ |
| $\phi_{\mathrm{spu}}$ | Spurious component of $\phi$ | Decomposition $\phi = \phi_{\mathrm{rob}} + \phi_{\mathrm{spu}}$ |
| $\gamma_{\mathrm{spu}}$ | Spurious-alignment term | $-\sum_{j \in S_k} W_{jk}(\phi_{\mathrm{spu}})_j \ge 0$ |
| $j_{\mathrm{rob}}$ | Index of a robust (never-gated) feature | $\tilde{E}_{j_{\mathrm{rob}} k} \ge \tau_k$ for all steps |

## D    DERIVATION OF ALIGNMENT ENERGY

Recall from Eq. (1)-(2) that the final classifier is linear in the representation,

$$z_k(x) = w_k^\top \phi_\theta(x) = \sum_{j=1}^{D} W_{jk}\, \phi_j(x), \qquad p_k(x) = \frac{\exp(z_k(x))}{\sum_{t=1}^{C} \exp(z_t(x))},$$

and that our per-neuron, per-class alignment score is defined as

$$e_{jk}(x) := -p_k(x)\, W_{jk}\, \phi_j(x). \tag{A.1}$$

In the main text we motivated $e_{jk}(x)$ as a confidence-weighted contribution of feature $j$ to class $k$. Here we provide a derivation based on the *free-energy* view of the classifier, working in parameter space.

**Free energy and sensitivity in weight space.**    Consider the (temperature-1) free energy of the model,

$$\mathcal{E}(x; W) = -\log \sum_{t=1}^{C} \exp\left(z_t(x)\right) = -\log \sum_{t=1}^{C} \exp\left(w_t^\top \phi_\theta(x)\right). \tag{A.2}$$

Differentiating $\mathcal{E}$ with respect to the logits yields

$$\frac{\partial \mathcal{E}(x; W)}{\partial z_k(x)} = -\frac{\exp(z_k(x))}{\sum_t \exp(z_t(x))} = -p_k(x). \tag{A.3}$$

Using the chain rule and the linearity of the logits, we obtain the gradient of the free energy with respect to a single weight $W_{jk}$:

$$\frac{\partial \mathcal{E}(x; W)}{\partial W_{jk}} = \frac{\partial \mathcal{E}}{\partial z_k(x)} \frac{\partial z_k(x)}{\partial W_{jk}} = \big(-p_k(x)\big)\,\phi_j(x). \tag{A.4}$$

Hence infinitesimal changes $\Delta W_{jk}$ yield the first-order perturbation

$$\Delta \mathcal{E}(x; W) \approx \sum_{j,k} \frac{\partial \mathcal{E}(x; W)}{\partial W_{jk}}\, \Delta W_{jk} = -\sum_{j,k} p_k(x)\,\phi_j(x)\,\Delta W_{jk}. \tag{A.5}$$

**Link-wise contribution and alignment energy.**   To isolate the contribution of a single link $(j, k)$, consider a straight-line interpolation in weight space from a reference value $\bar{W}_{jk}$ to the current value $W_{jk}$. A first-order approximation attributes to this link the energy change

$$\Delta \mathcal{E}_{jk}(x) \approx \frac{\partial \mathcal{E}(x; W)}{\partial W_{jk}}\,(W_{jk} - \bar{W}_{jk}). \tag{A.6}$$

Setting $\bar{W}_{jk} = 0$ for simplicity, and substituting Eq. (A.4), we obtain

$$\Delta \mathcal{E}_{jk}(x) \approx \big(-p_k(x)\,\phi_j(x)\big)\,W_{jk} = -p_k(x)\,W_{jk}\,\phi_j(x). \tag{A.7}$$

We therefore define the alignment score of link $(j, k)$ on input $x$ as this first-order contribution:

$$e_{jk}(x) \coloneqq \Delta \mathcal{E}_{jk}(x) = -p_k(x)\,W_{jk}\,\phi_j(x), \tag{A.8}$$

which recovers Eq. (2). This derivation provides a clear interpretation: $e_{jk}(x)$ *is the first-order change in free energy induced by moving the weight $W_{jk}$ from 0 to its current value, holding the representation $\phi_\theta(x)$ fixed.* More negative alignment corresponds to a stronger, confidence-weighted energy reduction along the link $(j, k)$. Finally, averaging over inputs with label $k$ yields the class-conditional alignment energy used in our method:

$$E_{jk} \coloneqq \mathbb{E}_{x \sim D_k}[e_{jk}(x)], \tag{A.9}$$

which is estimated while trining via an exponential moving average and used for percentile-gated suppression during training.

# E   PROOFS OF THE FOUR PROPERTIES OF AGS

We consider multi-class classification with penultimate representation $\phi_\theta(x) \in \mathbb{R}^D$, final linear layer $W = [w_1, \ldots, w_C] \in \mathbb{R}^{D \times C}$, logits $z(x) = W\phi_\theta(x)$ and probabilities $p_k(x) = \mathrm{softmax}_k(W\phi_\theta(x))$. For feature (neuron) $j$ and class $k$, the (per-example) *alignment score* is

$$e_{jk}(x) \triangleq -p_k(x)\,W_{jk}\,\phi_j(x),$$

and its class-$k$ energy is the population average $E_{jk} = \mathbb{E}_{x \sim \mathcal{D}_k}[e_{jk}(x)]$; in training we track an EMA $\widetilde{E}_{jk}^{(t)}$ of on-batch estimates (Eq. (5)). The classwise percentile threshold $\tau_k^{(t)}$ is the $q$-th percentile of $\{\widetilde{E}_{1k}^{(t)}, \ldots, \widetilde{E}_{Dk}^{(t)}\}$, and the binary gate is $s_{jk}^{(t)} = \mathbf{1}\{\widetilde{E}_{jk}^{(t)} < \tau_k^{(t)}\}$. The weight update (applied before backprop at step $t$) is

$$W_{jk} \leftarrow \left(1 - \alpha s_{jk}^{(t)}\right)\left(1 - \alpha c\right) W_{jk} \qquad \text{with } \alpha > 0,\ c = 0.05.$$

We will use the mild boundedness assumptions from §3.7:

$$\boxed{|\phi_j(x)| \leq R_\phi, \quad \|w_k\|_2 \leq R_w, \quad p_k(x) \in [0, 1]}.$$

**Property 1 (Confidence-weighted targeting and scale invariance).** *For $y = k$, $e_{jk}(x)$ becomes more negative as the alignment $W_{jk}\phi_j(x)$ increases while $p_k(x)$ does not decrease—thus higher-confidence, class-aligned contributions receive larger magnitude. Moreover, any logit-preserving rescaling $\phi' = a\phi$, $W' = W/a$ (with $a > 0$) leaves $W'^\top \phi' = W^\top \phi$ and hence $p' = p$; consequently $e'_{jk}(x) = e_{jk}(x)$, the ordering of $\{\widetilde{E}_{\cdot k}\}$, and the gated set are invariant. Lower-confidence samples (smaller $p_y$) produce alignment closer to 0 and thus exert less influence on gating.*

**Proof 1** *Fix an example with $y = k$. By definition $e_{jk}(x) = -p_k(x) W_{jk}\phi_j(x)$. If $W_{jk}\phi_j(x)$ increases and $p_k(x)$ does not decrease, then $-p_k(x) W_{jk}\phi_j(x)$ decreases (becomes more negative), i.e., $|e_{jk}(x)|$ increases in magnitude with the same (negative) sign. Hence, larger confidence $p_k(x)$ and stronger alignment $W_{jk}\phi_j(x)$ jointly push $e_{jk}(x)$ deeper into the negative tail.*

*For scale invariance, let $\phi' = a\phi$ and $W' = W/a$ with $a > 0$. Then for all classes $W'^\top \phi' = W^\top \phi$ so $p'_k(x) = p_k(x)$, and*

$$e'_{jk}(x) = -p'_k(x) W'_{jk} \phi'_j(x) = -p_k(x) \frac{W_{jk}}{a} (a\,\phi_j(x)) = e_{jk}(x).$$

*Therefore all $e'_{jk}(x)$ equal $e_{jk}(x)$ pointwise, their (EMA-smoothed) classwise energies $\widetilde{E}_{jk}$ have identical order statistics within each class $k$, and the percentile-gated set $\{(j,k) : \widetilde{E}_{jk} < \tau_k\}$ is unchanged. Finally, if $p_y$ is small, then $|e_{jy}(x)| = p_y|W_{jy}\phi_j(x)|$ is closer to 0, so low-confidence examples contribute less to the tail. This proves the property.*

> **Property 2 (Stable, budgeted gating via EMA and percentiles).** *The EMA update $\widetilde{E}^{(t)} = (1-\beta)\,\overline{E}^{(t)} + \beta\,\widetilde{E}^{(t-1)}$ (where $\beta \in [0,1)$ is the EMA parameter) is a convex combination with step-to-step drift bounded by $2(1-\beta)R_\phi R_w$. Per class $k$, at most $\lceil q\% \rceil$ features are gated (here $q$ is the gate budget in percent), because decisions depend only on the within-class order of $\{\widetilde{E}_{\cdot k}\}$, which is invariant under any strictly increasing transform. If $\widetilde{E}_{jk}^{(t-1)} \geq \tau_k^{(t-1)} + \Delta$ (gap $\Delta > 0$) and batch-induced perturbations have size at most $\delta$ (perturbation radius), then whenever*
> $$2(1-\beta)\delta < \Delta,$$
> *feature $j$ remains ungated at step $t$ (gate inertia). Larger mini-batches $B$ (with size $|B|$) and class-balanced sampling reduce the variance of $\overline{E}^{(t)}$ and hence the probability of flips.*

**Proof 2** *Because $\tilde{E}^{(t)} = (1-\beta)\bar{E}^{(t)} + \beta\tilde{E}^{(t-1)}$ is a convex combination, for each $(j,k)$,*

$$|\tilde{E}_{jk}^{(t)} - \tilde{E}_{jk}^{(t-1)}| = (1-\beta)|\bar{E}_{jk}^{(t)} - \tilde{E}_{jk}^{(t-1)}| \leq (1-\beta)\delta.$$

*Hence $|\tilde{E}_{jk}^{(t)} - \tilde{E}_{jk}^{(t-1)}| \leq 2(1-\beta)R_\phi R_w$, establishing the stated drift bound.*

*Budgeted gating by percentiles. By definition, $\tau_k^{(t)}$ is the $q$-th percentile of the multiset $\{\tilde{E}_{1k}^{(t)}, \ldots, \tilde{E}_{Dk}^{(t)}\}$. With deterministic tie-breaking, at most $\lceil q\% \rceil$ features are gated. Decisions are scale-free.*

*Gate inertia under bounded perturbations. Assume a per-class sup-norm batch perturbation radius $\delta$ so that for all $i$, $|\bar{E}_{ik}^{(t)} - \tilde{E}_{ik}^{(t-1)}| \leq \delta$. Then the entire vector $\tilde{E}_{\cdot k}^{(t)}$ deviates from $\tilde{E}_{\cdot k}^{(t-1)}$ by at most $(1-\beta)\delta$ in sup norm, and any order statistic (thus the percentile) shifts by at most $(1-\beta)\delta$.*

*Consequently, if at step $t-1$ we have a margin $\tilde{E}_{jk}^{(t-1)} - \tau_k^{(t-1)} \geq \Delta$, then at step $t$*

$$\tilde{E}_{jk}^{(t)} - \tau_k^{(t)} \geq \Delta - 2(1-\beta)\delta.$$

*Hence a sufficient condition for inertia is $\boxed{2(1-\beta)\delta < \Delta}$. Larger mini-batches $B$ and class-balanced sampling reduce the variance of $\bar{E}^{(t)}$ and hence the probability of flips. This concludes the proof.*

**Property 3 (Multiplicative decay contracts and sparsifies).** *Under the update $W_{jk} \leftarrow (1 - \alpha s_{jk})(1 - \alpha c) W_{jk}$, with $s_{jk} \in \{0, 1\}$ and $c = 0.05$, gated coordinates $(s_{jk} = 1)$ contract by $\rho = (1 - \alpha)(1 - \alpha c) < 1$ (the per-step contraction factor), yielding geometric attenuation across consecutive gated steps. For any threshold $\varepsilon > 0$, the number of gated entries with magnitude $> \varepsilon$ weakly decreases after one step (monotone sparsification). To first order in $\alpha$, the combined effect is equivalent to applying a class-conditional, feature-wise proximal shrinkage to gated links before the ERM update, plus a mild global decay to stabilize scale.*

**Proof 3** *For any $(j, k)$,*

$$|W_{jk}^+| = \left|(1 - \alpha s_{jk})(1 - \alpha c)\right| |W_{jk}| \quad \text{with} \quad 0 < (1 - \alpha)(1 - \alpha c) \leq (1 - \alpha c) < 1,$$

*hence each coordinate contracts (strictly if either $s_{jk} = 1$ or $c > 0$). In particular, a gated coordinate $(s_{jk} = 1)$ contracts by the factor $\rho = (1 - \alpha)(1 - \alpha c) \in (0, 1)$; iterating this over $\ell$ consecutive gated steps yields geometric decay $|W_{jk}^{(t+\ell)}| \leq \rho^\ell |W_{jk}^{(t)}|$.*

*For any fixed $\varepsilon > 0$, if $s_{jk} = 1$ and $|W_{jk}| > \varepsilon$ then after one step $|W_{jk}^+| = \rho|W_{jk}| < |W_{jk}|$; if $|W_{jk}| \leq \varepsilon$ it remains $\leq \varepsilon$. Thus the count of gated coordinates with magnitude $> \varepsilon$ cannot increase, establishing monotone sparsification on the gated subset. (The global factor $(1 - \alpha c) < 1$ only helps this conclusion.)*

*Finally, expanding to first order in $\alpha$ gives*

$$W_{jk}^+ = (1 - \alpha s_{jk})(1 - \alpha c) W_{jk} = \left(1 - \alpha(s_{jk} + c)\right) W_{jk} + O(\alpha^2),$$

*which is exactly the proximal map of the quadratic penalty $\frac{\alpha}{2}(s_{jk} + c) |W_{jk}|^2$ to first order (coordinate-wise shrinkage). This realizes a class-conditional, feature-wise proximal shrinkage on gated links $(s_{jk} = 1)$, plus a mild global shrinkage $c$ on all links, before the ERM gradient step, proving the property.*

**Property 4 (Bias suppression with preservation of robust features).** *When gated coordinates align with spurious components on worst-group examples, the pairwise margin*

$$m_{k \to t}(x) := (w_k - w_t)^\top \phi(x)$$

*obeys*

$$m_{k \to t}^+(x) \geq m_{k \to t}(x) + \alpha \gamma_{\mathrm{spu}} - \alpha c \left|(w_k - w_t)^\top \phi(x)\right|,$$

*where $S_k$ denotes the set of indices gated for class $k$, $\phi_{\mathrm{spu}}$ is the spurious component in the decomposition $\phi = \phi_{\mathrm{rob}} + \phi_{\mathrm{spu}}$, and $\gamma_{\mathrm{spu}} = -\sum_{j \in S_k} W_{jk} (\phi_{\mathrm{spu}})_j \geq 0$. This shows a net margin gain on worst-group inputs up to the small global decay term. Conversely, any feature whose smoothed alignment persistently remains above the classwise percentile, $\tilde{E}_{j_{\mathrm{rob}} k} \geq \tau_k$ (defining the index $j_{\mathrm{rob}}$ of a robust, never-gated feature), is deterministically never gated and is therefore preserved.*

**Proof 4** *Write $\phi = \phi_{\mathrm{rob}} + \phi_{\mathrm{spu}}$ and recall the per-class gated set $S_k = \{j : \tilde{E}_{jk} < \tau_k\}$. Under one AGS step,*

$$w_k^+ = (1 - \alpha c)\left(w_k - \alpha s_k \odot w_k\right), \qquad w_t^+ = (1 - \alpha c)\left(w_t - \alpha s_t \odot w_t\right),$$

*where $s_k$ (resp. $s_t$) is the 0/1 gate vector for class $k$ (resp. $t$) and "$\odot$" is the Hadamard product. Hence the updated margin satisfies*

$$m_{k \to t}^+(x) = (w_k^+ - w_t^+)^\top \phi = (1 - \alpha c)(w_k - w_t)^\top \phi - \alpha(1 - \alpha c)\left[(s_k \odot w_k)^\top \phi - (s_t \odot w_t)^\top \phi\right].$$

*For any real $u$, $(1 - \alpha c)u \geq u - \alpha c|u|$. On worst-group inputs the spurious part satisfies $(s_k \odot w_k)^\top \phi_{\mathrm{spu}} \leq 0$. Therefore the term contributes a positive margin gain when we define*

$$\gamma_{\mathrm{spu}} := -\sum_{j \in S_k} W_{jk}(\phi_{\mathrm{spu}})_j \geq 0.$$

*This yields $m^+_{k \to t}(x) \geq m_{k \to t}(x) + \alpha\gamma_{\mathrm{spu}} - \alpha c|m_{k \to t}(x)|$, showing a net margin gain on worst-group inputs up to the small global decay term.*

*Conversely, any feature whose smoothed alignment persistently remains above the classwise percentile, $\tilde{E}_{j_{\mathrm{rob}}k} \geq \tau_k$ (defining the index $j_{\mathrm{rob}}$ of a robust, never-gated feature), is deterministically never gated and is therefore preserved.*

**Notes on assumptions and why they are used.** We explicitly used the boundedness $|\phi_j| \leq R_\phi$, $|w_k|2 \leq R_w$, and $p_k \in [0, 1]$ to: (i) upper-bound the magnitude of any alignment $|ejk(x)| \leq R_\phi R_w$; (ii) control the EMA drift by $2(1 - \beta)R_\phi R_w$; and (iii) ensure percentile thresholds are well behaved. The *gate inertia* bound leverages a uniform per-batch perturbation radius $\delta$ to make a worst-case (yet clean) sup-norm argument; I provided a simple sufficient condition $2(1 - \beta)\delta < \Delta$ and then noted it implies the (looser) condition spelled out in the paper for the default $\beta$ range. Finally, Property 3's "proximal shrinkage" is made precise via the first-order equivalence to the proximal map of a quadratic penalty; this is the standard justification for multiplicative (coordinate-wise) shrinkage.

## DISCLOSURE OF LLM USAGE

We have made limited use of a large language model to support tasks such as refining wording, improving readability, and suggesting alternative formulations. All conceptual contributions, experimental design, implementation, and interpretation of results were carried out by the authors, who take full responsibility for the final content.

