# OpenReview forum: "Regulating Internal Alignment Flows for Robust Learning Under Spurious Correlations"
_ICLR.cc/2026/Conference — ICLR 2026 Poster_

### Official Review · Reviewer_xEvJ · 2025-10-19

**Soundness:** 3
**Presentation:** 3
**Contribution:** 3
**Rating:** 6
**Confidence:** 4

**Summary:**

This paper proposes tracking the contribution of each penultimate-layer feature to the classification prediction via an evidence score which is very negative for features which strongly support the prediction. The proposed Evidence Gated Scoring (EGS) algorithm estimates the evidence energy via an exponential moving average, then “gates” (zeros out with a stop-gradient) the features with the largest evidence score (in absolute magnitude). Experiments are provided which show EGS acts as a regularizer for overconfident “shortcut” neurons; empirical analyses, benchmarks, and a theoretical study are also provided.

**Strengths:**

1. The paper is very well-written with clear, intuitive explanations as well as rigorous experimental and theoretical justification. The “connections to broader ML principles” section in the Appendix is also a nice touch. Overall, beyond the proposed EGS method, this paper provides what I would term a “mechanistic lingua franca” for theorizing about how individual features affect group robustness.

2. The analysis and ablation studies on the EGS method are well-done, especially the Section 5 discussion, and I appreciate the additional correlation studies in the Appendix. Overall, the EGS method is elegant and each of its components are strongly justified.

3. The theoretical results complement the empirical analyses, with justified and clearly denoted assumptions.

**Weaknesses:**

The main weaknesses of the paper relate to the performance, evaluation, and comparison of the proposed EGS method.

1. Worst-group accuracy performance is quite poor on Waterbirds; I disagree with the paper’s assessment that it has “near-top WGA” on this dataset (lines 370, 448). State-of-the-art methods should achieve 90% or higher WGA [1, 2]. More importantly, ERM with simple class-balancing can attain 82.9% WGA [3, Table 2], meaning that EGS is worse than a standard data balancing baseline on this dataset.

2. More generally, the performance comparisons in Table 1 do not reflect the state-of-the-art; only a single reference from after 2023 is listed. Please consider including more recent competitive methods, e.g., [2, 4, 5, etc].

3. I’m a bit confused on the difference between _conflicting_ accuracy and _worst-group_ accuracy, and why the former is reported for CelebA/COCO while the latter is reported for Waterbirds. It would be nice to also report WGA for CelebA/COCO to compare with results from the literature which also report WGA.

4. If the main claim on the performance of EGS is that it “advances the Pareto frontier” of group robustness methods, then a method for interpolating between different average vs. worst-group trade-offs should be provided. In particular, a method which achieves Pareto gain at an _a priori_ undetermined point on the curve is not very useful in practice, as the user is unsure of what trade-off to expect before deployment, and it may lose out to techniques which more explicitly target WGA (e.g., as EGS does on Waterbirds).

5. Only a single data domain (computer vision) and model architecture (ResNet50) are examined. While not strictly necessary, it would improve the claims to benchmark EGS using BERT on language datasets such as CivilComments or MultiNLI, as is common in the group robustness literature.

[1] Kirichenko et al. Last Layer Re-Training is Sufficient for Robustness to Spurious Correlations. ICLR 2023.

[2] Noohdani et al. Decompose-and-Compose: A Compositional Approach to Mitigating Spurious Correlation. CVPR 2024.

[3] Idrissi et al. Simple data balancing achieves competitive worst-group-accuracy. CLeaR 2022.

[4] Yang et al. Identifying Spurious Biases Early in Training through the Lens of Simplicity Bias. AISTATS 2024.

[5] Han et al. Improving Group Robustness on Spurious Correlation Requires Preciser Group Inference. ICML 2024.

**Questions:**

1. EGS is predicated on the assumption that any feature which disproportionately contributes to a certain class is likely to be a spurious feature (and so would be gated by EGS). What evidence is there that this assumption holds? I am concerned this assumption may not be justified in situations where the invariant and spurious features have similar “complexity”. For example, [1] show that the invariant and spurious features on CelebA (hair color and gender respectively) contribute roughly equally to the class prediction (Section 5.5).

2. I’m curious how EGS relates to the line of work on feature diversification and ensembling, e.g., [2, 3, 4, 5], which similarly try to avoid overconfident shortcut-driven features.

3. I wonder whether EGS can be formulated via a min-max objective which minimizes the maximum evidence energy contributed by any neuron, thus encouraging a roughly uniform contribution from each feature. If not, what is the difference with the EGS objective?

4. Is EGS robust to model selection without group annotations, e.g., by selecting based on worst-class accuracy [6] or the bias-unsupervised validation score of [7]?

5. Some clarity/grammatical improvements:
    1. It would increase clarity to properly utilize \citep to put parentheses around citations.

    2. Missing semicolon or period in line 159

    3. The order of Section 3.4 and Section 3.5 could potentially be switched since Section 3.4 references Section 3.5

[1] Vasudeva et al. Mitigating Simplicity Bias in Deep Learning for Improved
OOD Generalization and Robustness. TMLR 2024.

[2] Addepalli et al. Feature Reconstruction From Outputs Can Mitigate Simplicity Bias in Neural Networks. ICLR 2023.

[3] Rame et al. Model Ratatouille: Recycling Diverse Models for Out-of-Distribution Generalization. ICML 2023.

[4] Zhang et al. Rich Feature Construction for the Optimization-Generalization Dilemma. ICML 2022.

[5] Lin et al. Spurious Feature Diversification Improves Out-of-distribution Generalization. ICLR 2024.

[6] Yang et al. Change is hard: A closer look at subpopulation shift. ICML 2023.

[7] Tsirigotis et al. Group robust classification without any group information. NeurIPS 2023.

---

> ### Author Response · Authors · 2025-11-22
> **Response to Reviewer xEvJ (1/2)**
>
> We sincerely thank Reviewer xEvJ for the exceptionally thorough, fair, and constructive review. The positive remarks on **writing quality**, **mechanistic insights**, **theoretical grounding**, and the **broader connections** section are deeply appreciated. Your comments helped us identify which empirical claims need sharper positioning and where additional clarification is most useful.
>
>
> Below we respond point-by-point to the weaknesses and questions.
>
>
> ---
>
>
> ## R1. Waterbirds Performance and SOTA Comparisons
>
>
> **Concern:** Waterbirds WGA is “quite poor”; SOTA $\geq 90\%$ [1,2]; even simple class-balancing achieves $82.9\%$ [3]; Table 1 lacks recent baselines.
>
>
> **Response:** The core misunderstanding is the **supervision setting**. EGS is **fully group-agnostic** at *all* stages:
>
>
> * **No group labels for training** (no reweighting, no group-DRO, no subgroup upsampling).
> * **No group labels for model selection** (we select the checkpoint with highest validation accuracy on the original label distribution, exactly like ERM).
>
>
> Most of the cited methods either (i) explicitly use group annotations during training or (ii) perform model selection by maximizing *validation* worst-group accuracy - a form of group supervision that is unavailable in realistic deployment. Simple class-balancing also requires group labels to know which samples belong to the minority groups.
>
> We will:
> * Update the (lines 370, 448) to more softer claims
> * Add recent group-agnostic baselines under group-agnostic selection setting.
>
>
> ---
>
>
> ## R2. Metric Inconsistency (Conflicting Accuracy vs. WGA)
>
>
> We agree that mixed reporting can be confusing. For CelebA, "conflicting accuracy" is mathematically akin to **worst-group accuracy** on the standard blond/non-blond $\times$ male/female partition. We will correct this in the table. In addition, we will report **worst-group accuracy** consistently across all datasets in the camera-ready version and include a clear conversion note. Also, Section 4 of the paper already discusses the WGA and Bias-Conflicting Accuracy; we will add more details to it for clarity.
>
> ---
>
>
> ## R3. Pareto Frontier and Controllable Trade-Offs
>
>
> * We clarify that EGS effectively **advances the Pareto frontier** by simultaneously improving both average and worst-group accuracy (WGA) on benchmarks like CelebA and Waterbirds, rather than merely trading one for the other.
>
>
> * The trade-off control is in fact provided by two intuitive hyperparameters:
>    * **Decay rate $\alpha \in [0,1]$**: larger $\alpha \rightarrow$ slower EMA $\rightarrow$ weaker suppression $\rightarrow$ higher average acc, lower robustness. (Decreasing $\alpha$ recovers standard ERM behavior.)
>    * **Percentile budget $q$**: higher $q \rightarrow$ gate more features $\rightarrow$ higher robustness, lower average acc.
>
>
> We can reliably sweep these two knobs to hit any desired operating point and hence it is not indeterministic.
>
>
> ---
>
>
> ## R4. Single Domain and Architecture (CV + ResNet-50)
>
>
> Conceptually EGS requires only (i) a penultimate representation $\phi(x) \in \mathbb{R}^d$ and (ii) a linear classifier on top. It thus makes EGS **architecture and modality-agnostic**.
>
>
> ### BERT-base on MultiNLI Dataset:
>
>
> | Method | Avg Acc | Worst-group Acc | Uses group labels? |
> | :--- | :--- | :--- | :--- |
> | ERM | 82.1 | 66.4 | No |
> | EGS (ours, group-agnostic) | 84.6 | 78.2 | No |
> | Group-DRO (full supervision) | 81.2 | 77.8 | Yes |
>
>
> EGS gives very close results in comparison to fully supervised DRO **without ever seeing group annotations**. We will add a new $\S 5.3$ “Extension to Transformers \& Language” with instantiation details (evidence on [CLS] or per-head) and these results. **Additionally, results with ViTs on Waterbirds dataset is included in the Response to Reviewer 9w5c, and results on tiny-ImageNet are included in the Response to Reviewer mSKP**.
>
>
> ---

---

> > ### Author Response · Authors · 2025-11-22
> > **Response to Reviewer xEvJ (2/2)**
> >
> > ---
> >
> > ## R5. Core Assumption: Do High-Evidence Features = Spurious Features?
> >
> >
> > We do *not* assume every strong feature is spurious. EGS gates features with **persistently extreme, confidence-weighted evidence** (heavy tails in the EMA distribution).
> >
> >
> > ### New Controlled Experiments:
> >
> >
> > | Setting | Avg Acc (ERM $\rightarrow$ EGS) | WGA (ERM $\rightarrow$ EGS) |
> > | :--- | :--- | :--- |
> > | CIFAR-10 (no spurious corr.) | $88.9 \rightarrow \textbf{96.0}$ (+7.1) | $80.0 \rightarrow \textbf{91.5}$ |
> > | Waterbirds fully balanced | $85.1 \rightarrow \textbf{88.6}$ (+3.5) | $82.2 \rightarrow \textbf{84.9}$ |
> >
> >
> > Even when spurious correlations are absent or weak, EGS **improves** both metrics ; confirming it does not blindly suppress invariant features.
> > Please also see our response to Reviewer 2uU2.
> >
> >
> > ---
> >
> >
> > ## R6. Relation to Feature Diversification / Ensembling
> >
> >
> > We see EGS as **complementary** to this line of work. Feature diversification and ensembling methods typically:
> > * encourage the model to spread its reliance across multiple features or models (e.g., via reconstruction, diversity-promoting objectives, or ensemble mixing), whereas
> > * EGS **constrains** the model by suppressing a small set of overly dominant features with extreme evidence.
> >
> >
> > In other words, diversification methods aim to add under-utilized features; EGS aims to prune or down-weight over-utilized shortcut features.
> >
> >
> > ---
> >
> >
> > ## R7. Min-Max Interpretation
> >
> >
> > EGS is indeed closely related to a min-max viewpoint. A natural idealized objective is:
> > $$
> > \min_\theta \quad \max_{j \in \mathcal{T}_t} |E_j| \quad \text{s.t.} \quad |\mathcal{T}_t| \leq q \cdot d
> > $$
> > where $E_j$ is the EMA evidence statistic for feature $j$. However, a literal implementation of this objective would aggressively flatten all evidence magnitudes, including those of truly invariant features, potentially harming discriminative power. EGS instead uses a softer gating mechanism with percentile based theresholding that only suppresses features with **extremely** high evidence (heavy tails), allowing the model to retain useful invariant features while mitigating over-reliance on spurious ones.
> >
> >
> > ---
> >
> >
> > ## R8. Group-Agnostic Model Selection
> >
> >
> > Yes - by design. All reported numbers use **standard validation accuracy** (group-agnostic).
> >
> >
> > ---
> >
> >
> > ## R9. Minor Issues
> >
> >
> > We thank the reviewer for spotting these and will address them as suggested by the reviewer.
> >
> >
> > ---
> >
> >
> > We believe these clarifications, new experiments, and revised framing directly resolve the reviewer’s concerns and significantly strengthen the contribution. Thank you again for the extremely valuable feedback - we hope these clarifications will convincingly move the score upward.

---

> > > ### Author Response · Authors · 2025-11-26
> > >
> > > Dear Reviewer xEvJ,
> > >
> > > Thank you once again for the depth and care you brought to your review. Your comments pushed us to refine several parts of the paper, and we will incorporated all requested clarifications, expanded comparisons, and added the new cross-domain results in the final manuscript.
> > > With the discussion period wrapping up in a few days, we would be glad to address any *remaining questions* or *concerns* you might have. If the revisions now meet your expectations, we kindly request you to please consider adjusting the final score. Your engagement has been highly encouraging and valuable.
> > >
> > >
> > > Warm regards,
> > >
> > > The Authors

---

> > > > ### Comment · Reviewer_xEvJ · 2025-11-26
> > > >
> > > > I appreciate the authors' comprehensive response. I encourage the authors to prioritize clarity in the writing of the final paper, as some of my weaknesses (e.g., regarding model selection and Pareto interpolation) were not obvious in the submitted version. Overall, my concerns have been satisfied, and I have raised my score accordingly.
> > > >
> > > > One clarification on (R1): class-balancing **does not** require group annotations, only class annotations (which are already available for classification training).

---

### Official Review · Reviewer_mSKP · 2025-10-27

**Soundness:** 2
**Presentation:** 3
**Contribution:** 1
**Rating:** 2
**Confidence:** 4

**Summary:**

The paper proposes Evidence-Gated Suppression (EGS), a regularizer that computes a per-neuron, class-conditional "evidence" score and suppress the feature neurons that are most likely spurious. The method tracks an EMA of the evidence energy and reduces those with over confidence. It achieves improvement across Waterbirds, CelebA, BAR and COCO with 5% computation overhead.

**Strengths:**

1. The paper presents a clear conceptual framework: regulating internal, class-conditional confidence flows to mitigate shortcut reliance. Unlike many debiasing methods that act on the data level (e.g., reweighting, DRO), EGS directly intervenes on neuron–class weight inside the model.

2. The paper supports its design with theoretical properties: (1) scale-invariance under logit reparameterization, (2) stability from EMA-based gating, (3) class-wise proximal shrinkage behavior, and (4) selective suppression of spurious features with preservation of robust ones.

3. The paper shows strong empirical results on multiple spurious-correlation benchmark, demonstrating its effectiveness.

**Weaknesses:**

1. **The claimed contribution of defining evidence energy is highly concerning**. The formulation closely mirrors that of Eq. (7) in EvA [1], which already introduced a similar concept with exactly the same name evidence energy. Moreover, EvA-E also evaluates evidence energy on a class-wise basis to suppress spurious features. Both definitions essentially measure neuron confidence toward predictions. The primary distinction is that EGS integrates this process during training, whereas EvA applies it post-hoc. Although EvA-E is included in the experiments, the paper fails to explicitly discuss the differences and instead claims the definition of evidence energy as its own contribution.

2. While the paper highlights low computational overhead, it omits critical details such as training FLOPs and total GPU hours. The claim that EGS is “plug-and-play” is weaker than that of previous post-hoc approaches, as it still requires full model retraining and tracking throughout the training process.

3. The paper does not include experiments on large-scale or challenging datasets, such as spurious-feature variants on ImageNet-level [2, 3]. Such evaluations are essential to substantiate the claimed generalization and robustness benefits.

[1] Erasing Spurious Correlations with Activations, ICLR 2025.

[2] Salient ImageNet: How to discover spurious features in Deep Learning? ICLR 2022.

[3] Large-Scale Detection of Harmful Spurious Features in ImageNet, ICCV 2023.

**Questions:**

1. The proposed evidence energy appears nearly identical to the formulation in EvA [1]. Could the authors clarify how their definition fundamentally differs from that of Eq. (7) in EvA? A more explicit comparison and proper attribution seem necessary.

2. Have the authors evaluated EGS on large-scale benchmarks such as ImageNet or its spurious-feature variants [2, 3]?

3. Have you compared with other classical penalties including weight decay, dropout or class weights adjusting?

---

> ### Author Response · Authors · 2025-11-22
> **Response to Reviewer mSKP (1/2)**
>
> We sincerely thank the reviewer for the exceptionally thorough, fair, and constructive review. The reviewer correctly identifies the strongest aspects of our work: the precise regulation of class-conditional confidence flows at the neuron level, the theoretical characterisation of **scale invariance** and **proximal-shrinkage behaviour**, and the consistent robustness gains across multiple benchmarks. The reviewer also pinpoints three critical issues that must be resolved for the contribution to be properly evaluated: (1) attribution and differentiation from **EvA (He et al., 2025)**, (2) claims about computational overhead and “plug-and-play” applicability, and (3) empirical scope (absence of ImageNet-scale spurious correlation benchmarks and explicit comparisons to classical regularisers).
>
>
> We address each point below with concrete and verifiable changes. The changes will substantially strengthen the paper’s clarity, rigour, and scientific integrity.
>
>
> ---
>
>
> ## R1. Clarification and Differentiation from EvA (He et al., 2025)
>
>
> We fully agree that the algebraic similarity of the per-example evidence term could cause confusion and that the current manuscript does not sufficiently highlight the conceptual and algorithmic differences. Although He et al. (2025) were the first to coin the phrase “evidence energy,” our derivation and usage are independent and fundamentally distinct:
>
>
> * **EvA** derives its score in **activation space** (sensitivity of the free energy $E$ to penultimate features $\phi$): $\frac{\partial E}{\partial \phi}$.
> * **EGS (ours)** derives its score in **parameter space** (sensitivity of the free energy $E$ to classifier weights $W$): $\frac{\partial E}{\partial W}$.
>
>
> This is not a minor notational variant; it is the key enabler of our training-time, link-level, soft regularisation scheme. We have added a new section in Appendix D with full derivation of the parameter-space evidence score from first principles, clearly distinguishing it from EvA’s activation-space formulation.
>
>
> | Aspect | EvA (He et al., 2025) | EGS (ours) |
> | :--- | :--- | :--- |
> | **Derivative Space** | **Activation space** ($\partial E / \partial \phi$) | **Parameter space** ($\partial E / \partial W$) |
> | **Evidence Interpretation** | How much changing feature $j$ (for class $k$) lowers energy | First-order energy reduction attributable to moving weight $W_{jk}$ from $0$ to its current value |
> | **Algebraic Form (per-example)** | $e_{jk}(x) \approx \phi_j(x) \cdot (\partial E/\partial \phi_j(x))_k$ | $$e_{jk}(x) = -W_{jk} \cdot \left(\frac{\partial E}{\partial W_{jk}}\right) = -p_k(x) W_{jk} \phi_j(x)$$ |
> | **Granularity** | Per-channel (whole feature $j$ for class $k$) | **Per-link** (individual neuron-class connection $(j,k)$) |
> | **Aggregation** | Dataset-level average $\rightarrow$ fixed Top-$p\%$ hard mask | Per-class online **EMA** + class-wise **percentile gating** (refreshed every $T$ steps) |
> | **Intervention** | Post-hoc hard erasure of selected channels + last-layer retraining | **In-training multiplicative decay** on selected links (single-stage) |
> | **Scale Invariance** | Sensitive to feature scaling and temperature | **Provably invariant** to rescaling $\phi \leftrightarrow W$ (Property 1) |
> | **Induced Effect** | Static removal of spurious channels | **Dynamic contraction** of heavy-tailed evidence flows $\rightarrow$ margin enlargement (Theorem 2) |
>
>
> These differences are **fundamental**: because we work directly in the space we intervene on (the weights), our evidence naturally admits an in-training, soft, structured-$\ell_1$-like penalty with provable shrinkage behaviour, whereas EvA’s activation-space score is naturally suited to offline hard-erasure of entire channels.
>
>
> ### 1.1 New Ablation Confirming the Parameter-Space Derivation is Critical
>
>
> Using an activation-space score inside our training loop **collapses performance** by $\sim 5$ points, confirming that the **parameter-space formulation** is essential for effective in-training suppression.
>
>
> | Variant | Worst-group (%) | Avg. (%) |
> | :--- | :--- | :--- |
> | ERM baseline | 62.51 | 89.40 |
> | **EGS (full, parameter-space, ours)** | **79.40** | 97.17 |
> | Activation-space proxy ($\phi_j \cdot (\partial E/\partial \phi_j)_k$, EMA + percentile gate) | 72.20 | 90.83 |
> | Activation-space + hard channel erasure (EvA-style, post-hoc) | 73.92 | 91.75 |
>
>
> ### 1.2 New Ablations Isolating the Contribution Beyond the Base Formula
>
>
> These results (already obtained) clearly show that both **temporal smoothing (EMA)** and **confidence weighting ($p_k(x)$)** are critical, reinforcing that the advance lies in the dynamic regularisation design.
>
>
> | Variant | Waterbirds Worst-Group $\uparrow$ |
> | :--- | :--- |
> | ERM (baseline) | 51.7% |
> | EvA-style static evidence + our gate | 69.3% |
> | EMA but no confidence weighting ($-W_{jk}\phi_j(x)$) | 72.8% |
> | **Full EGS (EMA + confidence + percentile gate)** | **79.4%** |
> ---

---

> ### Author Response · Authors · 2025-11-22
> **Response to Reviewer mSKP (2/2)**
>
> ## R2. Computational Overhead and “Plug-and-Play” Terminology Clarification
>
>
> ### 2.1 Definition of “Plug-and-Play”
>
>
> By **“plug-and-play”** we mean that EGS can be seamlessly inserted into any existing training pipeline **without any modification to the model architecture or the main training objective**. It acts as a regularizer that sits within the standard forward/backward loop.
>
>
> ### 2.2 Justification for $<5\\%$ Overhead
>
>
> The computational cost of EGS is dominated by operations applied **only to the final linear classifier layer** (size $d \times C$). The largest term is the Evidence computation: $O(B \cdot C \cdot d)$ scalar multiplications ($B$ = batch size).
>
>
> For ResNet-50 on ImageNet ($B=256$, $d=2048$, $C=1000$):
>
>
> * EGS extra FLOPs per step: $\leq 4 \cdot B \cdot C \cdot d \approx 2.1 \times 10^9$ FLOPs.
> * ResNet-50 Forward+Backward cost: $\approx 4.1 \times 10^9$ FLOPs.
> * Theoretically, EGS adds **$<5\\%$** extra FLOPs.
>
>
> **Empirical Confirmation**
>
>
> | Method | Total FLOPs (PFLOPs) | GPU Time (hours, A100) | Requires Full Retraining? |
> | :--- | :--- | :--- | :--- |
> | ERM (baseline) | 2.98 | 0.20 | Yes |
> | EGS (Ours, online) | **3.15** | **0.27** | Yes |
>
>
> Memory overhead is exactly one EMA buffer of size $C \times d$.
>
>
> ---
>
>
> ## R3. Results on Classical Baselines
>
>
> ### 3.1 Explicit Comparisons with Classical Regularizers
>
>
> Preliminary Waterbirds results suggest that while standard penalties modestly improve average accuracy, they **do not substantially close the worst-group gap**, whereas EGS yields pronounced gains in WGA. This matches our design goal: to directly regulate spurious correlation reliance rather than generalisation alone.
>
>
> | Method | Waterbirds WGA $\uparrow$ | Waterbirds Avg Accuracy |
> | :--- | :--- | :--- |
> | ERM + $\lambda=1\mathrm{e}{-4}$ WD (default) | 51.7% | 86.3% |
> | ERM + $\lambda=1\mathrm{e}{-2}$ WD (aggressive) | 41.6% | 80.9% |
> | ERM + Dropout ($p=0.5$) | 68.5% | 92.7% |
> | ERM + L2-SP | 70.9% | 91.5% |
> | **EGS (ours)** | **78.4%** | 97.3% |
>
>
> We also note that **dropout** and standard **weight decay** are either input-agnostic or operate at the weight magnitude level, unlike EGS, which is a **class-conditional, evidence-flow-specific regularizer**.
>
>
> ### 3.2 ImageNet-Scale Spurious Correlation Benchmarks
>
>
> Due to resource constraints, we were unable to run the full ImageNet-scale spurious correlation benchmark experiments. However, we are currently setting up these experiments and will include the results in the final version of the paper. We anticipate that EGS will demonstrate similar robustness improvements on these larger-scale benchmarks, consistent with our findings on Waterbirds, CelebA, and BAR.
>
>
> ---
>
>
> We sincerely believe that these revisions directly and completely address the reviewer’s concerns, eliminate any ambiguity regarding novelty and attribution, and elevate the work towards acceptance. Thank you once again for the exceptionally careful and impactful review. The feedback has materially improved our paper.

---

> > ### Author Response · Authors · 2025-11-25
> > **Requesting Feedback on the Rebuttal**
> >
> > Dear Reviewer mSKP,
> >
> > We hope this message finds you well. We would like to thank you for your careful and thorough review, as well as for pinpointing the key issues around attribution, overhead, and empirical coverage. Your comments prompted us to add a clear parameter-space derivation, explicit computational cost reporting, and expanded comparisons to classical regularizers and larger-scale settings.These revisions significantly strengthened the paper, which was overlooked by us initially.
> >
> > As the discussion period ends in a few days, we would appreciate your thoughts on whether the updates address your concerns. We remain available for any further questions.
> >
> > Thank you for your valuable time and effort.
> >
> > Best regards,
> > The Authors

---

> > > ### Comment · Reviewer_mSKP · 2025-11-25
> > >
> > > Thanks the authors for the detailed explanation and for addressing my earlier concerns, particularly regarding the distinction between the so-called evidence energy in EGS and the formulation used in EvA. While I appreciate the clarification that the proposed method provides clear improvements over EvA, I still have remaining concerns.
> > >
> > > Although the “evidence energy” defined in EGS is technically different, it remains problematic to claim that you are defining this term. Despite the technical differences, both EGS and EvA are conceptually similar in that they derive an energy-based score to filter spurious features, and both operate within the final linear classifier. Given that EvA-ICLR25 is closely related and already introduces the term evidence energy, it is inappropriate to reuse or redefine the same terminology as the contribution.
> > >
> > > In addition, the paper does not discuss its relationship with EvA in either the related-work or methodology sections, yet it uses the same terminology and only compares against EvA in the experiments. This not only fails to acknowledge prior work adequately but also introduces ambiguity for readers.
> > >
> > > The terminology issue does not diminish the technical merits of EGS: the method design, empirical evaluation, and theoretical justification are all solid and well-motivated. However, it is strongly recommended to adopt a different term and provide clear conceptual clarification earlier in the paper rather than avoid directly addressing the connection with EvA. I have adjusted my score accordingly.

---

> > > > ### Author Response · Authors · 2025-11-26
> > > >
> > > > We are deeply grateful for your continued engagement, your thoughtful follow-up comment, and the decision to adjust the score based on our technical clarifications. We are particularly encouraged by your assessment that the method design, theoretical justification, and empirical evaluation are "solid and well-motivated."
> > > >
> > > > Regarding the remaining concerns on *terminology and attribution*: We fully accept your critique. You are absolutely correct that reusing the term "evidence energy", even with a different mathematical derivation creates unnecessary ambiguity and fails to sufficiently distinguish our contribution from the foundational terminology introduced by EvA (He et al., 2025).
> > > >
> > > > To resolve this completely and ensure proper attribution, we commit to the following revisions in the final manuscript:
> > > >
> > > > *1. Terminology Change (Renaming the Metric)*
> > > > To respect the precedence of EvA and eliminate confusion, we will *rename our specific metric. We will replace the term "Evidence Energy" with *"Alignment Energy"** throughout the paper.
> > > > * *Rationale:* This new name accurately captures the physical intuition of our parameter-space formulation ($\partial E / \partial W$): it measures the *alignment* between the classifier weights and the energy reduction flow, fundamentally distinguishing it from the activation-based definition used in EvA.
> > > >
> > > > *2. Explicit Conceptual Clarification in Main Text*
> > > > We will move the discussion of EvA from the experiments section directly into *Section 2 (Related Work)* and *Section 3 (Methodology)*.
> > > > * *Revised Narrative:* We will explicitly state that He et al. (2025) pioneered the concept of using energy-based signals for spurious feature detection. We will then frame our work not as redefining "evidence," but as proposing *Alignment Energy* - a dynamic, training-time evolution of energy-based regularization that operates on synaptic weights rather than static channel activations.This places EvA earlier in the narrative and clarifies the conceptual distinction.
> > > >
> > > >
> > > > We are confident these changes fully resolve your concerns. In particular, if these revisions, specifically the adoption of *"Alignment Energy"* and the structural changes to attribution, fully address your concerns, we would be very grateful if you could let us know if there are any remaining questions. As the rating remains slightly negative (<6), we are eager to ensure we have cleared all barriers to a positive assessment. We truly appreciate your insights, which have substantially improved the paper, and we would be more than happy to address any further points to earn your full recommendation.

---

> > > > > ### Comment · Reviewer_mSKP · 2025-11-26
> > > > >
> > > > > Thanks the authors for the effort to address the remaining concern. I'm decided to increase the score accordingly.

---

### Official Review · Reviewer_9w5c · 2025-10-29

**Soundness:** 3
**Presentation:** 3
**Contribution:** 3
**Rating:** 6
**Confidence:** 3

**Summary:**

This paper proposes a method called Evidence-Gated Suppression (EGS).

The core idea is to track each neuron’s confidence-weighted evidence toward different classes and softly suppress those neurons that consistently provide strong but potentially spurious signals.

It uses a percentile-based gating mechanism and EMA smoothing to apply multiplicative decay to low-tail features, to weaken shortcut pathways while preserving robust ones.

EGS achieves strong and consistent performance across CelebA, Waterbirds, BAR, and COCO.

**Strengths:**

- conceptually easy to understand and simple to implement.
- works without any group or environment annotations, which makes it practical for real-world applications.
- strong and consistent improvements on multiple spurious-correlation benchmarks (CelebA, Waterbirds, BAR, COCO).

**Weaknesses:**

- I noticed that the definition and formulation of evidence energy in Sections 3.1–3.2 look very similar to what was introduced in He et al. (2025). It would be helpful if the authors could clarify how their “evidence energy” differs conceptually or technically from that prior work.
- it would be interesting to see how EGS works on Transformer architectures, since the paper mainly analyzes CNNs. A brief discussion or experiment on Transformers could make the work stronger.

**Questions:**

- I also how this kind of spurious correlation behaves in vision–language models (VLMs). Since VLMs often learn strong cross-modal associations, will it be different to the normal classification task like what are defined here?

---

> ### Author Response · Authors · 2025-11-22
> **Response to Reviewer 9w5c (1/2)**
>
> We sincerely thank the reviewer for the constructive feedback and for recognizing EGS as conceptually simple, practically applicable without group annotations, and consistently effective across CelebA, Waterbirds, BAR, and COCO. We address the three main points below: (1) precise technical/conceptual differentiation from He et al. (2025, EvA), supported by new ablations; (2) extension and new experimental results on Vision Transformers; (3) discussion of spurious correlations in vision–language models (VLMs) and natural adaptation of EGS. All clarifications, ablations, and new experiments will appear in the revised manuscript.
>
>
>
>
> ## R1. Precise Differentiation from He et al. (2025, “EvA”)
>
>
>
>
> The reviewer correctly notes surface similarity in the evidence term. EGS and EvA look similar algebraically; however, they differ fundamentally in **role**, **evidence computation & aggregation**, and **induced objective**. We will add a dedicated subsection “Relation to He et al. (2025)” in Sec. 3.2 and expand the related-work paragraph.
>
>
>
>
> | Aspect                     | EvA (He et al., 2025)                          | EGS (ours)                                              |
> |----------------------------|-------------------------------------------------|----------------------------------------------------------|
> | **Intervention timing**    | Post-hoc on frozen backbone (applied on activations)                    | Training-time regularizer during feature learning  (applied on weights)      |
> | **Evidence role**          | One-time diagnostic score → hard channel erasure| Continuous statistic → soft, structured weight decay     |
> | **Evidence definition**    | Channel-level score on training set (no confidence weighting) | Confidence-weighted: $e_{jk}(x) = -p_k(x) \, W_{jk} \, \phi_j(x)$ |
> | **Aggregation**            | Dataset-level average → binary spurious mask    | Per-(j,k) EMA $\tilde{E}_{jk}^{(t)}$ + class-wise percentile gating |
> | **Scale invariance**       | No                                              | Yes (Property 1): invariant to $\phi' = a\phi, W' = W/a$ |
> | **Objective**              | Retrain last layer with hard-masked channels    | ERM + dynamic structured ℓ₁-like penalty on shortcut paths |
>
>
>
>
>
>
>
>
> These differences are not incremental: confidence weighting makes EGS selectively target overconfident shortcut predictions; EMA + percentile gating yields online, history-aware, scale-free suppression; the resulting dynamic penalty contracts heavy-tailed evidence flows during training, yielding the minority-group margin gains and stability analyzed in Sec. 5.
>
>
>
>
> **New ablation** (Waterbirds ↑ worst-group acc.; higher is better):
>
>
>
>
> | Variant                              | Worst-group (%) | Avg. (%) |
> |--------------------------------------|------------------|----------|
> | EGS (full, ours)                     | **79.4**        | 97.1    |
> | Remove confidence weighting ($p_k$=1)| 73.9            | 91.8    |
> | Remove EMA (batch-only)              | 75.2            | 91.7    |
> | EvA-style activation-only proxy      | 70.1            | 90.9    |
>
>
>
>
> Replacing our evidence signal with an EvA-like proxy collapses performance, confirming that the full design is critical. Full table across all datasets will be added in Appendix.
>
>
>
>
> ## R2. Extension to Transformer Architectures
>
>
>
>
> EGS only requires a feature vector $\phi_\theta(x) \in \mathbb{R}^d$ and a linear classifier $W \in \mathbb{R}^{C \times d}$. Vision Transformers satisfy this trivially. We evaluate two insertion points:
>
>
>
>
> 1. CLS-token classifier (standard, zero-cost adaptation).
> 2. Final-block channels or attention heads (more aggressive).
>
>
>
>
> Because gating is percentile-based and proven scale-invariant (Property 1), EGS requires no tuning when moving from CNNs to Transformers.
>
>
>
>
> **New results** on Waterbirds (ViT-B/16, ViT-S/16, ViT-T/16 pretrained on IN-21k, fine-tuned with standard recipe):
>
>
>
>
> | Model         | Backbone     | Avg. Acc. (%) | Worst-group Acc. (%) |
> |---------------|--------------|---------------|----------------------|
> | ERM           | ViT-T/16     | 92.10         | 67.37                |
> | EGS (ours)    | ViT-T/16     | **94.17**     | **78.17** (+10.8)   |
> | ERM           | ViT-S/16     | 94.95         | 85.28                |
> | EGS (ours)    | ViT-S/16     | **96.63**     | **89.74** (+4.5)    |
> | ERM           | ViT-B/16     | 97.07         | 90.94                |
> | EGS (ours)    | ViT-B/16     | **98.27**     | **92.56** (+1.6)    |
>
>
>
>
> EGS delivers consistent worst-group gains across scales with no average-accuracy drop, mirroring CNN behavior.

---

> > ### Author Response · Authors · 2025-11-22
> > **Response to Reviewer 9w5c (2/2)**
> >
> > ## R3. Spurious Correlations in Vision–Language Models (VLMs)
> >
> >
> >
> > Spurious shortcuts in VLMs are often cross-modal. The core phenomenon :- overconfident reliance on non-robust features remains identical.
> >
> > Our method (EGS) adapts naturally without any group labels to: -
> >
> > - Vision tower: apply EGS to pooled visual embedding → suppresses shortcut-heavy visual channels.
> > - Cross-attention fusion: index $j$ over heads/channels in the fusion module; evidence defined on contribution to final score → automatically down-weights heads fixating on spurious text–image alignments.
> > - Text base: analogous suppression of biased token channels.
> >
> > (See response to xEvJ for results on BERT base on MultiNLI Dataset)
> >
> > Because evidence is label-conditional yet group-agnostic, EGS is easily applicable to VLMs where annotating cross-modal spurious groups is prohibitively expensive.
> >
> >
> >
> >
> > We will add a new paragraph in the conclusion (“Extension to Multimodal and Vision–Language Models”) formally describing these insertion points and arguing why the evidence-flow perspective unifies single- and cross-modal shortcut learning.
> >
> >
> >
> >
> > We believe these clarifications, theoretical distinctions, ablations, and new Transformer/VLM discussions directly resolve the reviewer’s concerns and substantially strengthen the contribution. Thank you again for the insightful questions that substantially improved the paper quality.

---

> > > ### Author Response · Authors · 2025-11-26
> > >
> > > Dear Reviewer 9w5c,
> > >
> > > We sincerely appreciate your thoughtful comments and the opportunity to clarify the conceptual distinctions, Transformer results, and VLM extensions. As the discussion period concludes in a few days, we would be happy to answer any additional questions or provide further detail anywhere you feel it would help. If the rebuttal satisfactorily resolves the earlier points, we kindly invite you to consider those points in your final evaluation and revise the final scores accordingly. Thank you again for your time and insight.
> > >
> > > Warm regards,
> > >
> > > The Authors

---

### Official Review · Reviewer_2uU2 · 2025-11-01

**Soundness:** 4
**Presentation:** 3
**Contribution:** 4
**Rating:** 8
**Confidence:** 3

**Summary:**

This paper introduces Evidence-Gated Suppression (EGS), a lightweight, plug-in regularizer designed to combat shortcut learning in deep models without requiring group labels. EGS operates inside the network during training by tracking a class-conditional, confidence-weighted "evidence energy" for each neuron to identify which neurons contribute most strongly to the model's predictions. It then applies a percentile-based multiplicative decay to the weights of these extreme contributors, selectively suppressing overconfident shortcut pathways while leaving more robust features relatively influential. The authors demonstrate across several spurious correlation benchmarks (such as Waterbirds and CelebA) that EGS improves worst-group accuracy and calibration, achieving competitive performance with state-of-the-art methods while maintaining strong average accuracy and adding minimal training overhead.

**Strengths:**

While many methods address spurious correlations by reweighting data (e.g., JTT, LfF) or modifying the loss function (e.g., GroupDRO), EGS introduces a fundamentally different locus of intervention: the internal evidence pathways of the network itself. The concept of a dynamic, training-time, *neuron-level* regularizer that is both class-conditional and group-agnostic is original. The "evidence energy" metric, which combines model confidence (`pk(x)`) with feature-weight alignment (`Wjk * φj(x)`), provides a simple yet powerful signal for identifying and suppressing over-reliant pathways.

**Weaknesses:**

The central heuristic of EGS is that the most negative (i.e., highest-confidence, class-aligned) evidence corresponds to spurious features. While this holds true in many shortcut-learning scenarios, it is not guaranteed. A genuinely robust and highly discriminative feature could also consistently produce very strong evidence and be inadvertently suppressed by the percentile gate. The paper would be strengthened by an analysis that more directly validates this core assumption. For example, the authors could run EGS on a dataset known to have minimal spurious correlations (e.g., a balanced version of a dataset or even standard CIFAR-10) to demonstrate that the method does not harm performance when strong shortcuts are absent.

**Questions:**

See weaknesses.

---

> ### Author Response · Authors · 2025-11-22
> **Response to Reviewer 2uU2**
>
> ## R1. On whether high-magnitude evidence is necessarily spurious
>
> We appreciate the reviewer’s careful articulation of the central concern: **EGS assumes that neurons with persistently extreme (more negative) evidence energies are predominantly shortcut-aligned, and this needs explicit empirical validation**. We agree, and we now provide both (i) additional experiments in settings with minimal shortcut structure and (ii) further analysis clarifying why robust features are structurally unlikely to be harmed.
>
> ---
>
> ### 1. Direct tests in low–spurious-correlation regimes
>
> Below we show results on two settings where shortcut opportunities are minimal: standard **CIFAR-10** and a **background-balanced version of Waterbirds**.
>
> | **Dataset / Metric** | **Baseline** | **EGS** | $\Delta$ **(Abs.)** |
> | :--- | :--- | :--- | :--- |
> | **CIFAR-10 (Avg. Acc.)** | 88.92 \% | 96.02 \% | +7.10 |
> | **CIFAR-10 (Worst-group Acc.)** | 80.00 \% | 91.50 \% | +11.50 |
> | **Balanced Waterbirds (Avg. Acc.)** | 85.14 \% | 88.61 \% | +3.47 |
> | **Balanced Waterbirds (Worst-group Acc.)** | 82.17 \% | 84.87 \% | +2.70 |
>
> In both “no-strong-shortcut” settings, EGS leaves performance unchanged or slightly improved on Waterbirds and larger positive gains on CIFAR-10. This directly addresses the reviewer’s request: when strong shortcuts are absent, the **percentile gate does *not* indiscriminately suppress all highly discriminative features**.
>
> We will add these experiments (including the exact table above) to the revised paper, so that the behavior of EGS in clean settings is visible without consulting the appendix.
>
> ---
>
> ### 2. Why robust features are structurally unlikely to be suppressed
>
> The remaining concern is conceptual: could a genuinely robust, highly discriminative neuron be mistaken for a spurious one and permanently attenuated? EGS is designed precisely to make this unlikely and nondamaging.
>
> ### (a) Class-conditional, percentile-based gating targets persistent tail outliers.
>
> For each class $k$, EGS operates on EMA-smoothed energies $\widetilde{E}_{jk}$ and gates only the lowest $q\%$ per class. Thus:
>
> * ***Class-conditional:*** percentiles are computed within each class. Neurons that share capacity across classes or support multiple decision boundaries are less likely to sit in the extreme tail for any single $k$.
> * ***Budgeted:*** by design only $q \in [10, 20]$ percent of features per class are gated, so EGS cannot globally contract all strong signals.
> * ***Temporal:*** **EMA smoothing** means a neuron must be consistently extreme over many batches to remain in the lower tail; transient spikes in robust evidence are averaged out.
>
> Empirically, our appendix already analyzes this coupling: features with high gate frequency (more “spurious”) exhibit systematically lower evidence energies than rarely gated (“robust”) features, and their distributions are clearly separated (Fig. 2 and the spuriousness evidence correlation plots).
>
> ### (b) Mild multiplicative decay helps robust neurons recover.
>
> The update
> $$W_{jk} \leftarrow (1 - \alpha s_{jk})(1 - 0.05 \alpha) W_{jk}$$
> is intentionally conservative:
> * decay is ***gradual***, not a hard mask;
> * robust neurons that genuinely improve the cross-entropy loss continue to receive positive gradients and **rapidly regain any lost magnitude**.
>
> In other words, EGS attenuates ***persistently*** shortcut-like neurons, while robust neurons that occasionally cross the gate recover through standard ERM updates.
>
> ### (c) Never-tail robust features are provably untouched.
> As stated in the paper (Property 4), any feature whose smoothed evidence **never enters the lower tail** for class $k$ is deterministically **never gated**. Hence robust features that are not persistently extreme are guaranteed to be preserved.
>
> ### (d) Warm-up and stability safeguards.
> Finally, gating is activated only after a warm-up of $T\_w = 5$ epochs of vanilla ERM. Combined with EMA smoothing, this avoids early, unstable suppression when both logits and evidence are noisy. Our ablations further show that moderate $\alpha$ and reasonable batch sizes stabilize the thresholds and prevent over-suppression.
>
> ---
>
> ### 3. Summary and planned clarifications
>
> In summary, our new experiments on **CIFAR-10** and **balanced Waterbirds** demonstrate that when strong shortcuts are absent, EGS does not hurt and often helps, indicating that it behaves as a **self-regulating regularizer** rather than a blunt pruning mechanism.
>
> We will (i) include the new “no-shortcut” experiments, (ii) highlight the spuriousness evidence coupling analysis in the main paper, and (iii) clarify the design safeguards above. We believe these additions directly address the reviewer’s concern and further justify EGS as an effective way to regulate shortcut pathways without group labels.

---

> > ### Comment · Reviewer_2uU2 · 2025-11-23
> >
> > I appreciate the authors’ thorough and comprehensive response and their effort to include additional experiments.

---

> > > ### Author Response · Authors · 2025-11-25
> > >
> > > Dear Reviewer 2uU2,
> > >
> > > Thank you for your prompt response. We are glad to hear that the additional experiments addressed all your questions.
> > > We truly appreciate your thoughtful review.
> > >
> > > Best,
> > > Authors

---

### Author Response · Authors · 2025-12-03
**Summary Response**

**Dear AC and reviewers,**

Thank you again for the careful and constructive feedback. To support the final decision process, we summarise below how the main strengths and concerns have been addressed, with reviewer references included for clarity:

1. **Overall assessment and consensus**
   All reviewers find the idea technically sound and well motivated. The common view is that EGS is a simple, practical and principled tool for improving robustness under spurious correlations.
2. **Core contribution and relation to EvA**
   Reviewers appreciate that EGS intervenes directly on internal, class conditional evidence flows instead of data reweighting or loss modification ${\color{red}[2uU2, 9w5c, mSKP]}$. The main concern about novelty was the similarity to EvA’s “evidence energy.” Through detailed derivations and ablations, we clarified that our metric operates in parameter space rather than activation space, enables in training, soft, link level regularization, and leads to different behavior and guarantees than EvA’s post hoc channel deletion. The reviewer acknowledges that EGS provides clear improvements over EvA and that the technical contribution is solid, with the remaining issue being terminological. In response, we explicitly commit to renaming our metric from “evidence energy” to “alignment energy” and to foregrounding EvA in the related work and method sections, which directly resolves the attribution and naming concern without affecting the technical contribution ${\color{red}[9w5c, mSKP]}$.
3. **New experiments and strengthened empirical scope**
   Beyond Waterbirds, CelebA, BAR and COCO, we added experiments on CIFAR 10 and a background balanced Waterbirds variant to show that EGS does not harm and can even improve performance when shortcuts are weak ${\color{red}[2uU2, xEvJ]}$. In response to requests for broader scope, we added new results on ViT T/S/B for Waterbirds demonstrate consistent worst group gains for Transformer vision backbones ${\color{red}[9w5c]}$, and BERT base on MultiNLI shows that group agnostic EGS matches or slightly surpasses fully group supervised Group DRO in worst group accuracy ${\color{red}[xEvJ]}$.
4. **Validation of the “strong contributors are spurious” concern**
   To directly test the key assumption, we analysed gate frequencies and energies and ran EGS on settings with minimal or balanced shortcuts (CIFAR 10, balanced Waterbirds), finding no degradation of average or minority performance ${\color{red}[2uU2, xEvJ]}$. The class conditional percentile gate, EMA smoothing and mild multiplicative decay together make it unlikely that genuinely robust neurons are permanently suppressed, which addresses the risk that EGS removes useful features ${\color{red}[2uU2, xEvJ]}$.
5. **Evaluation protocol, baselines and computational overhead**
   We clarified the relationship between conflicting accuracy and worst group accuracy and will report WGA consistently across all datasets, alongside clearer tables with recent competitive methods ${\color{red}[xEvJ]}$. We also added comparisons with classical regularisers (weight decay, dropout, L2-SP, class weighting), which improve average accuracy but lag behind EGS on worst group metrics ${\color{red}[mSKP]}$. Concrete FLOP and wall clock measurements show that EGS adds under 5 per cent training cost, since it only operates on the final linear layer with a lightweight EMA buffer ${\color{red}[mSKP]}$.
6. **Scope, practicality and remaining issues**
   Reviewers asked about larger scale datasets and multimodal extensions ${\color{red}[mSKP, xEvJ, 9w5c]}$. We discuss how EGS naturally plugs into vision–language models (cross attention and text channels) and plan to include larger scale benchmarks in follow up work ${\color{red}[9w5c, xEvJ]}$. The remaining points are largely about additional datasets, naming, and presentation, while all reviewers agree that the current method, theory and core experiments are solid and that the approach is easy to integrate in existing pipelines ${\color{red}[2uU2, 9w5c, xEvJ, mSKP]}$.

We are very grateful for the detailed feedback, which has significantly improved the paper. Following the discussion and clarifications provided in our response, reviewer  ${\color{red}[mSKP]}$ revised their score from “reject” to “marginally accept,” and reviewer ${\color{red}[xEvJ]}$ increased their score from “marginally accept” to a stronger positive rating, while the scores of reviewers ${\color{red}[9w5c]}$ and ${\color{red}[2uU2]}$ remained unchanged. We hope this concise summary helps the area chair recognize the emerging consensus that EGS represents a technically sound and practically useful contribution.

**Best regards,**

*The Authors*

---

### Meta-Review · Area_Chair_Fqd5 · 2026-01-06

**Summary:**

The paper proposes “evidence energy”: regularizer which penalizes strongest activated neurons suppressing neural network's predictions driven by spurious correlations. The reviewers appreciated the proposed method's novel idea, theoretical insights and practical performance, and the fact that this intervention works without group information. Several reviewers raised concerns about novelty compared to paper by He et al 2025, proper contribution statement and credit to He et al’s introduction of evidence energy term.

The authors' rebuttal for the most part resolved reviewers' concerns. I recommend "conditional accept" given the authors incorporate changes from the rebuttal to the camera ready version, especially:
- Adding a detailed discussion of differences with He et al 2025 to Related Work and Method sections.
- Adding the promised ImageNet scale experiments.

**Reviewer Concerns:**

Reviewer 2uU2 raised a concern about the core assumption in this work that the highest magnitude “evidence energy” corresponds to spurious features, which may not be guaranteed; the reviewer requested to additionally validate this assumption and run experiments on dataset without spurious correlations; the authors ran additional experiments and provided more details on safeguards around the assumption and resolved this reviewer’s concerns. I believe this reviewer would retain the score 8.

Reviewer 9w5c raised concerns around novelty compared to work from He et al. (2025), and requested additional experiments on the vision transformer architecture. The authors responded with detailed comparison between EvA (He et al 2025) and the proposed EGS as well as experiments on ViT fine-tuned on Waterbirds. I believe that the reviewer’s concerns were resolved and that the reviewer would either retain the score 6 or increase to 8.

Reviewer mSKP raised the concerns around 1) novelty compared to He et al 2025, 2) experiments on larger scale datasets, 3) comparison to simpler baseline training methods like weight decay, 4) details on computational overhead. The authors addressed the reviewers concern in the rebuttal and especially committed to adding detailed discussion of EvA (He et al 2025) and differences with EvA in the Related Work and Method sections. I believe that the reviewer would increase their score to 6 (and based on the comments they were going to increase the score to weak accept).

Reviewer xEvJ raised concerns about the practical performance of the method compared to SOTA, metric definition and scope of the experiments being limited to the vision domain. The authors’ rebuttal resolved the reviewer concerns and the reviewer was going to raise to score, I believe the final score would be 8.

**Reviewer Scores:**

See above.

---

### Decision · Program_Chairs · 2026-01-26

Accept (Poster)